# Fructose reprogrammes glutamine-dependent oxidative metabolism to support LPS-induced inflammation

Nicholas Jones [1,7], Julianna Blagih[2,7], Fabio Zani[2], April Rees[1], David G. Hill[3], Benjamin J. Jenkins[1], Caroline J. Bull [3,4], Diana Moreira[5], Azari I. M. Bantan[1], James G. Cronin [1], Daniele Avancini[6], Gareth W. Jones [3], David K. Finlay[5], Karen H. Vousden [2], Emma E. Vincent [3,4,8 ✉] & Catherine A. Thornton[1,8 ✉]

Fructose intake has increased substantially throughout the developed world and is associated with obesity, type 2 diabetes and non-alcoholic fatty liver disease. Currently, our understanding of the metabolic and mechanistic implications for immune cells, such as monocytes and macrophages, exposed to elevated levels of dietary fructose is limited. Here, we show that fructose reprograms cellular metabolic pathways to favour glutaminolysis and oxidative metabolism, which are required to support increased inflammatory cytokine production in both LPS-treated human monocytes and mouse macrophages. A fructose-dependent increase in mTORC1 activity drives translation of pro-inflammatory cytokines in response to LPS. LPS-stimulated monocytes treated with fructose rely heavily on oxidative metabolism and have reduced flexibility in response to both glycolytic and mitochondrial inhibition, suggesting glycolysis and oxidative metabolism are inextricably coupled in these cells. The physiological implications of fructose exposure are demonstrated in a model of LPS-induced systemic inflammation, with mice exposed to fructose having increased levels of circulating IL-1β after LPS challenge. Taken together, our work underpins a pro-inflammatory role for dietary fructose in LPS-stimulated mononuclear phagocytes which occurs at the expense of metabolic flexibility.

[1] Institute of Life Science, Swansea University Medical School, Swansea University, Swansea, UK. [2] The Francis Crick Institute, London, UK. [3] Cellular and Molecular Medicine, University of Bristol, Bristol, UK. [4] MRC Integrative Epidemiology Unit, University of Bristol, Bristol, UK. [5] School of Biochemistry and Immunology, Trinity Biomedical Sciences Institute, Trinity College Dublin, Dublin, Ireland. [6] San Raffaele Telethon Institute for Gene Therapy, San Raffaele Scientific Institute, Milan, Italy. [7] These authors contributed equally: Nicholas Jones, Julianna Blagih. [8] These authors jointly supervised this work: Emma E. Vincent, Catherine A. Thornton. ✉email: emma.vincent@bristol.ac.uk; c.a.thornton@swansea.ac.uk

Typically, activation of the human innate immune system requires the rewiring of cellular metabolic pathways largely to favour glucose metabolism[1–5]. However, in the various nutrient environments they inhabit, monocytes will be exposed to a range of different carbon sources, the availability of which will likely dictate their metabolism and phenotype. One such carbon source is fructose, the second most abundant dietary sugar found in humans. Fructose is metabolised by glycolysis either by keto-hexokinase producing fructose-1-phosphate, a substrate for aldolase B[6] (in the liver, kidneys and intestines for example) or converted to the glycolytic intermediate fructose-6-phosphate by hexokinase (HK), albeit at a lower rate than glucose[7,8].

Fructose intake has increased substantially throughout the Western world, largely attributed to elevated sucrose and high fructose corn syrup consumption[9] and is thought to exacerbate various non-communicable conditions such as obesity, type 2 diabetes and non-alcoholic fatty liver disease[9]. Chronic fructose consumption in these conditions has recently been shown to drive hepatic fructolysis, where the expression of lipogenic genes is enhanced[10–12].

Typically, physiological levels of fructose in the circulation range from 0.04 to 0.2 mM[13]; however, there are several pathophysiological scenarios in which levels of fructose are elevated. For example, peripheral blood levels can exceed 1 mM in patients with haematological malignancies such as acute myeloid leukaemia and acute lymphoblastic leukaemia[14]. In addition, fructose concentrations in the bone marrow microenvironment of haematological cancer patients can reach up to 5 mM[14]. Alterations in the glucose to fructose ratio, particularly when glucose is scarce, enables acute myeloid leukaemia blasts to significantly enhance fructose uptake[14]. Localised mouse tissue microenvironments, such as the liver, kidneys and jejunum, also have elevated levels of fructose metabolism[15]. Therefore, there are various pathophysiological scenarios and tissue microenvironments where monocytes will be exposed to either equimolar concentrations of fructose and glucose or concentrations of fructose exceeding that of glucose.

The impact of elevated fructose exposure on the immune system has not been investigated extensively. Chronic fructose exposure in rats results in a more inflammatory phenotype of bone marrow mononuclear cells[16]. While there is some evidence that lipopolysaccharide (LPS)-stimulated human dendritic cells are able to produce enhanced levels of pro-inflammatory cytokines when cultured in fructose as opposed to glucose, the underlying metabolic rewiring that enables this pro-inflammatory phenotype has not been investigated[17].

Here we characterise how human monocytes and mouse macrophages respond metabolically and functionally to fructose exposure. We show that activated mononuclear phagocytes demonstrate plasticity in engaging metabolism of this alternative carbon source, yet it leaves the cells metabolically inflexible and vulnerable to further metabolic challenge. Fructose exposure reprogrammes cellular pathways to favour glutaminolysis and oxidative metabolism, which support an inflammatory phenotype in both human and mouse mononuclear phagocytes. Finally, we demonstrate that a short-term high fructose diet promotes inflammation in vivo, suggesting that our findings have pathophysiological significance.

## Results

**Fructose exposure promotes an oxidative phenotype**. We sought to investigate the metabolic response to fructose exposure in activated human monocytes in comparison to other monosaccharides (glucose and galactose) or complete sugar withdrawal. Galactose has been used previously to promote oxidative phosphorylation (OXPHOS) in T cells by reducing glycolysis[18]. Using the Seahorse Bioanalyzer, we initially injected the monosaccharide (glucose, fructose or galactose) or media control containing no sugar. After 1 h, monocytes were stimulated with LPS and the corresponding extracellular acidification rate (ECAR) and oxygen consumption rate (OCR) were used to measure glycolysis and OXPHOS, respectively, for the duration.

Monocytes incubated with glucose demonstrated a robust increase in basal and LPS-induced ECAR (Fig. 1A). By contrast, monocytes treated with fructose or galactose had low baseline levels of glycolysis, only slightly greater than no sugar controls (Fig. 1A). In addition, monocytes treated with fructose, galactose or no sugar had only a modest increase in ECAR post-LPS exposure (Fig. 1A). Upon activation, glucose-treated cells reduced their OCR, consistent with the reported metabolic switch from OXPHOS to glycolysis upon activation[3]. Monocytes treated with fructose, galactose or no sugar had an initial burst of increased oxygen consumption and maintained higher OCR for the duration of the assay (Fig. 1B). This demonstrates metabolic flexibility of human monocytes, in this case towards OXPHOS when the cells are unable to engage in glycolysis.

After the introduction of 2-deoxy-D-glucose (2-DG), ECAR was reduced in all conditions (Fig. 1C). OCR was increased in glucose-treated monocytes, a compensatory response to glycolysis inhibition, whereas in fructose, galactose or no sugar treatment a decrease in OCR was observed (Fig. 1D). These data suggest that LPS-stimulated monocytes treated with fructose or galactose direct pyruvate towards the mitochondrial tricarboxylic acid (TCA) cycle for OXPHOS, whereas glucose treatment directs pyruvate towards lactate production. We confirmed extracellular levels of lactate are indeed reduced in fructose compared to glucose-treated monocytes (Supplementary Fig. 1A). This divergence appears to account for the differences in OXPHOS rates observed. Overall, LPS-stimulated human monocytes treated with fructose maintained an elevated oxidative phenotype with low ECAR, akin to galactose or no sugar treatment, whereas glucose availability preferentially maintained elevated levels of glycolysis at the expense of reduced oxygen consumption (Fig. 1E).

In agreement with a previous study, LPS-stimulated monocytes cultured for 24 h with galactose or no sugar had a significant reduction in viability in comparison to the glucose-treated control (Fig. 1F)[19]. However, we observed no difference in viability between glucose and fructose treatment (Fig. 1F).

To further elaborate fructose metabolism in LPS-stimulated monocytes, we incubated them with either uniformly labelled $^{13}C_6$-fructose or $^{13}C_6$-glucose and performed stable isotope tracer analysis (SITA). Activated monocytes were able to transport fructose into the cell (presumably via GLUT5 expression—Supplementary Fig. 1B) and incorporated comparable levels of carbon into intracellular lactate while increasing incorporation into TCA cycle intermediates and amino acids in comparison to glucose (Supplementary Fig. 1C–E). These results suggest that the cells have the ability to metabolise fructose carbon and use it in the TCA cycle.

Collectively, these data reveal that fructose treatment promotes a low glycolytic rate without compromising cell viability. This demonstrates the metabolic flexibility of human monocytes as well as their ability to utilise an alternative carbon source.

**Glycolysis and OXPHOS are coupled in fructose-treated human monocytes**. To confirm that the observed increase in OCR in fructose-treated monocytes was due to OXPHOS as opposed to an increase in other oxygen-consuming processes, we utilised the ATP synthase inhibitor, oligomycin. Here, monocytes were treated with either glucose or fructose and allowed to rest prior to LPS exposure. Oligomycin was later injected and the

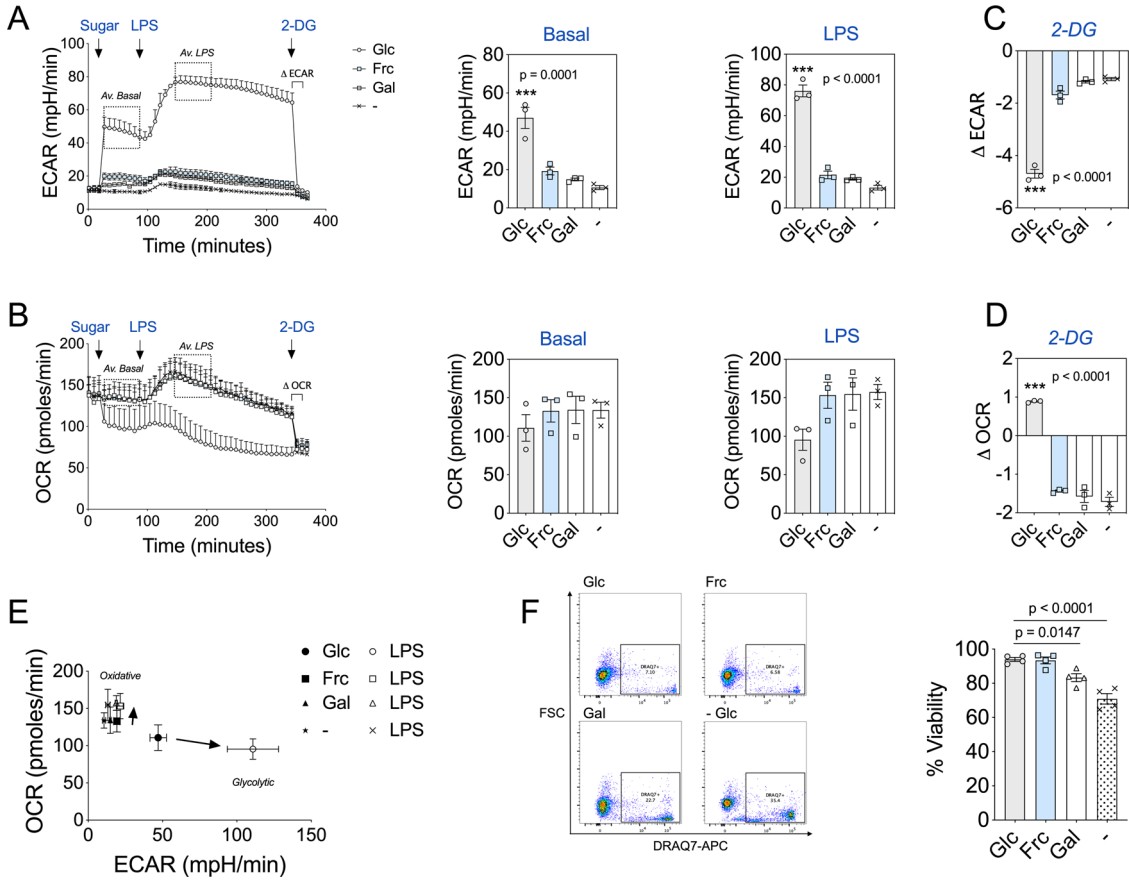

**Fig. 1 Fructose mimics a metabolic profile akin to nutrient restriction. A** Seahorse extracellular flux analysis of ECAR in monocytes before and following injections of glucose, fructose, galactose (11.1 mM) or no sugar, LPS (10 ng/mL) and 2-DG (100 mM) at the time points indicated with average basal (*Av. basal*) and LPS-stimulated (*Av. LPS*) ECAR values. **B** Analysis of OCR in monocytes with the same injections as in panel (**A**), including average basal (*Av. basal*) and LPS-stimulated (*Av. LPS*) OCR values. **C** Change in ECAR (Δ ECAR) post 2-DG treatment. **D** Change in OCR (Δ OCR) post 2-DG treatment. **E** OCR versus ECAR map of average basal and LPS-stimulated values for glucose, fructose, galactose and no sugar treatment groups. Arrows indicate the shift in metabolism from average basal to average LPS. **F** Representative flow cytometry plot and DRAQ7 viability measurements of glucose, fructose, galactose and no sugar monocytes cultured for 24 h with LPS (10 ng/mL). Statistical significance was assessed using a one-way ANOVA with Dunnett's (**A**–**D**) or Tukey's (**F**) multiple comparisons test. Data are either representative of either three (**A**–**E**) or four independent experiments (**F**). Data are expressed as mean ± SEM; ***$p \leq 0.001$. Source data are provided as a Source Data file.

bioenergetic changes were monitored over time. While OCR of both glucose or fructose LPS-simulated monocytes decreased to the same level upon oligomycin treatment (Fig. 2A), the drastic reduction of OCR in fructose-cultured monocytes reflects a greater reliance on OXPHOS. Cells reliant on OXPHOS may demonstrate metabolic flexibility upon oligomycin treatment by increasing glycolysis. However, surprisingly, ECAR in fructose-cultured monocytes decreased in response to oligomycin, suggesting the lack of metabolic adaptation in these cells (Fig. 2B, C).

Secondly, to establish whether the elevated ECAR levels post-LPS treatment reflected glycolytic activity as opposed to other acidifying processes, we used a lactate dehydrogenase inhibitor (GSK2837808A; LDHi). Here, the increased ECAR in response to LPS stimulation was reduced in glucose-treated monocytes upon LDHi treatment (Fig. 2D). By contrast, LDHi modestly impacted ECAR in fructose-treated cells, arguing that fructose-mediated glycolysis is coupled to OXPHOS. The low level of ECAR under this condition is most likely due to acidification of the media by an alternative source to lactate (Fig. 2D). We confirmed this was not due to changes in LDH phosphorylation in fructose- versus glucose-treated cells (Supplementary Fig. 1F).

The corresponding OCR levels increased in the glucose-treated monocytes upon LDHi treatment as pyruvate is directed towards

OXPHOS, again demonstrating the bioenergetic flexibility of human monocytes. By contrast, OCR in fructose-treated monocytes remained unchanged for the duration of the assay (Fig. 2E, F). These data further demonstrate that glycolysis and OXPHOS are tightly coupled in fructose-treated cells, revealing impaired metabolic flexibility in comparison to glucose-treated cells (Fig. 2C, F).

**Fructose treatment enhances LPS-induced inflammation.** Given the distinct metabolic characteristics of human monocytes exposed to fructose (Fig. 1A), we were intrigued to investigate the impact on monocyte function. Fructose-treated monocytes produced elevated levels of a panel of secreted cytokines, namely interleukin-1β (IL-1β), IL-6, IL-8, IL-10 and tumour necrosis factor (TNF) (Fig. 3A–E), with IL-1β, IL-8, IL-10 and TNF reaching statistical significance in comparison to glucose treatment. Despite elevated levels of cytokine secretion, there were no differences in various surface markers associated with monocyte activation (HLA-DR, CD80, CD86, CD62L, CCR5 and CCR2) between the glucose- or fructose-treated monocytes (Supplementary Fig. 2A).

To determine whether the increased production of cytokines was a consequence of increased transcription, we performed

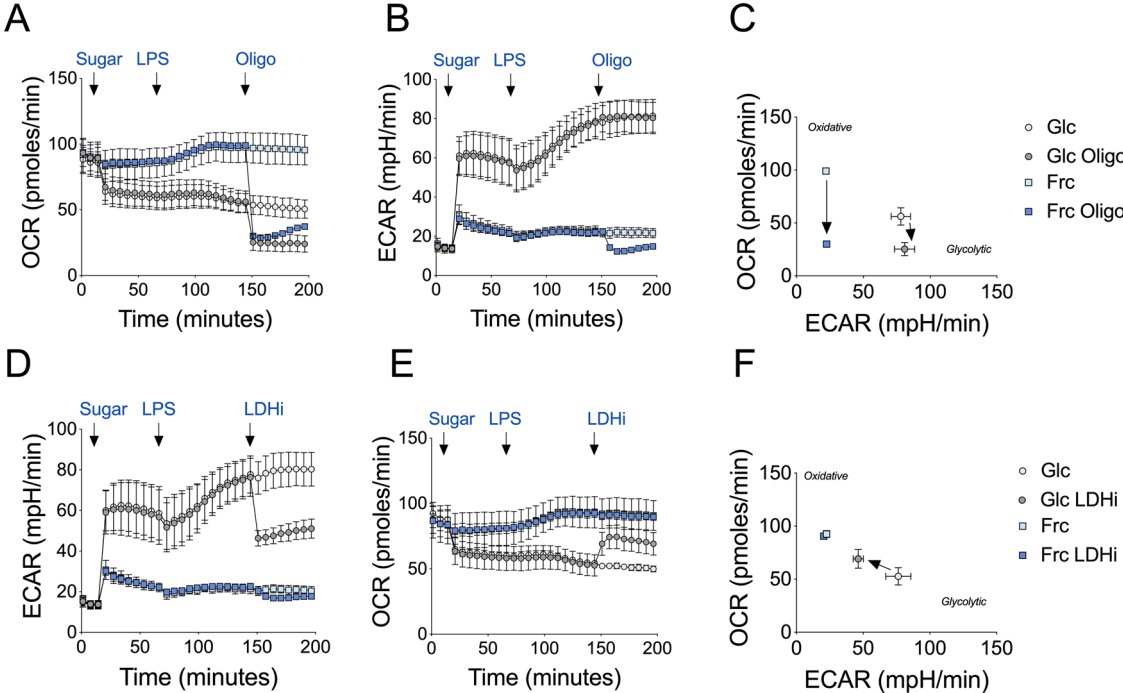

**Fig. 2 Fructose treatment enhances oxidative metabolism.** Seahorse extracellular flux analysis of **A** OCR and **B** ECAR in monocytes before and following injections of glucose or fructose (both 11.1 mM), LPS (10 ng/mL) and oligomycin (1 μM) at the time points indicated. **C** OCR versus ECAR map of average of a single value pre- and post-oligomycin injection for glucose- and fructose-treated monocytes. Arrows indicate the shift in metabolism upon inhibitor exposure. Analysis of **D** ECAR and **E** OCR in monocytes as **A** with a final injection of a lactate dehydrogenase inhibitor (GSK2837808A, LDHi; 10 μM). **F** OCR versus ECAR map as **C** with pre- and post- LDHi injection. Data are representative of three independent experiments and are expressed as mean ± SEM. Source data are provided as a Source Data file.

RNA-sequencing (RNA-seq) analysis of LPS-stimulated mono-cytes treated with glucose or fructose. Surprisingly, we observed only five genes that were significantly changed between the two conditions, with no alteration to transcript levels for the cytokines of interest (Supplementary Fig. 2B). Consistent with this, the expression level of genes encoding cytokines was also comparable in monocytes treated with either glucose or fructose (Fig. 3F), suggesting that the increased cytokine production was not through transcriptional regulation. Fructose has recently been reported to activate mTORC1 via dihydroxyacetone phosphate sensing[20]. Consistent with this, phosphorylation of the down-stream mTORC1 target, S6 ribosomal protein, was elevated significantly in LPS-stimulated monocytes treated with fructose (Fig. 3G). Of note, we observed no differences in the induction of AMPK (AMP-activated protein kinase) signalling between the two groups (Supplementary Fig. 2C). Collectively, these observa-tions demonstrate that LPS stimulation in the presence of fructose enhances inflammatory cytokine production, in part, through increased mTORC1-mediated translation.

**Fructose treatment drives a sustained oxidative phenotype.** LPS-stimulated monocytes in the presence of fructose clearly enhance oxygen consumption in the short term (Fig. 1C). Next, we wanted to determine whether heightened oxygen consump-tion was sustained long term in activated monocytes. Here, using the Seahorse Bioanalyzer, we performed a mitochondrial stress test with a final injection of the ionophore, monensin[21], following a 24-h incubation period with either glucose or fructose. Elevated levels of oxygen consumption were indeed sustained long term in fructose-treated LPS-stimulated monocytes (Fig. 4A). This was characterised by increased levels of oxidative parameters such as basal respiration, ATP-linked respiration and a higher percentage

of coupling efficiency in comparison to glucose (Fig. 4B–D and Supplementary Fig. 3A–D). Corresponding ECAR levels were unsurprisingly higher in glucose-treated monocytes in compar-ison to fructose-treated cells (Fig. 4E, F and Supplementary Fig. 3E). Notably, fructose-treated monocytes had a significantly higher ratio of basal OCR/ECAR ratio in comparison to glucose, indicating their commitment to oxidative metabolism (Fig. 4G).

We next assessed the contributions of glycolysis derived- and OXPHOS-derived ATP production[21]. These analyses show glucose-treated monocytes as glycolytic cells functioning at their maximal glycolytic ATP production rate at baseline (Fig. 4H). At maximal bioenergetic capacity, they demonstrate flexibility towards ATP production from OXPHOS, but still derive the majority of their ATP from glycolysis and therefore remain classed as glycolytic cells (Fig. 4H and Supplementary Fig. 3F, G). By contrast, fructose-treated monocytes are oxidative (deriving the majority of their ATP from OXPHOS) at baseline and this phenotype is further exacerbated when cells are at maximal bioenergetic capacity (Fig. 4H and Supplementary Fig. 3F, G). They do, however, have some metabolic scope to increase ATP production from glycolysis (Fig. 4H).

We reasoned that differential mitochondrial properties might explain the heightened oxidative phenotype of fructose-treated monocytes. However, we observed no difference in mitochondrial content (MitoTracker Green), membrane potential (tetramethylr-hodamine ethyl ester) or mitochondrial-derived reactive oxygen species (ROS) (Fig. 4I). Further to this, we also assessed levels of the individual respiratory complexes by immunoblot. Again, we observed no difference in the abundance of complexes I–IV between fructose- or glucose-treated monocytes (Fig. 4J). A high fructose diet has been shown to increase de novo lipogenesis in the liver[10]. This correlates with increased mitochondrial ATP produc-tion, which may support this energy-demanding process[22,23].

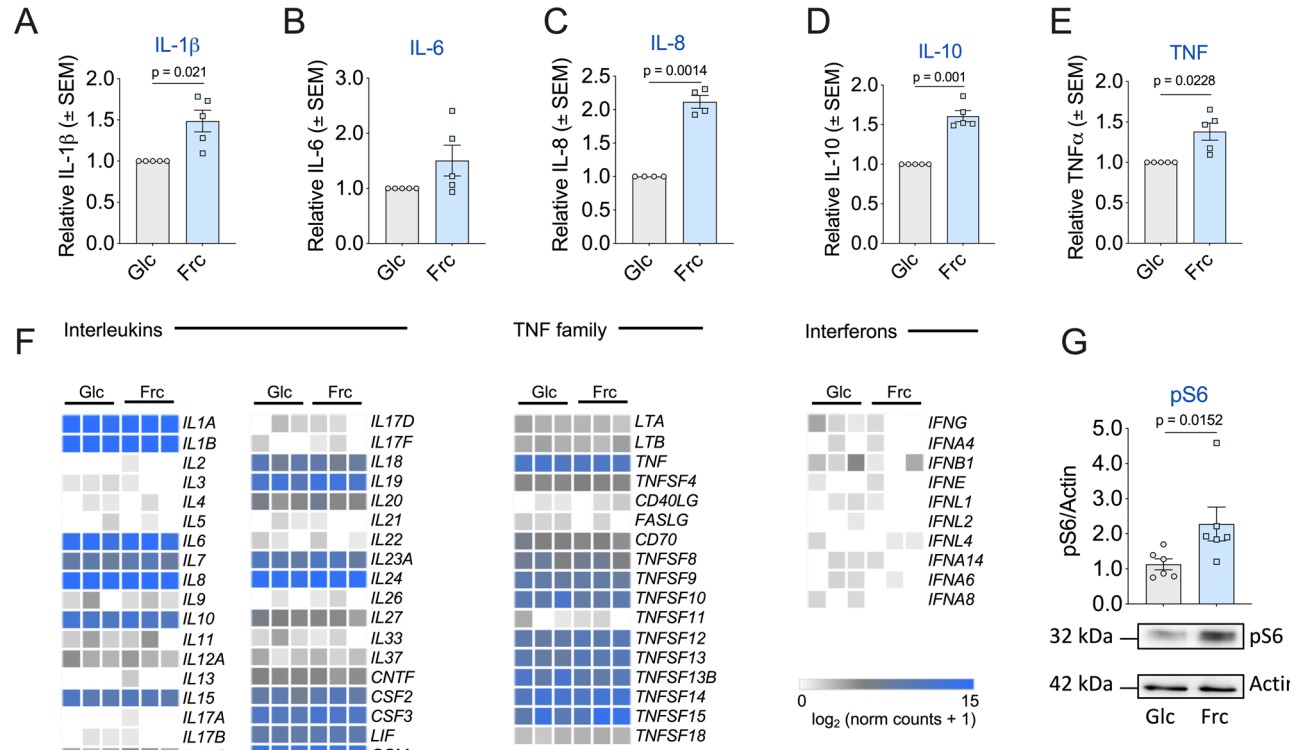

**Fig. 3 Fructose promotes a more inflammatory phenotype.** Extracellular cytokine release of **A** IL-1β, **B** IL-6, **C** IL-8, **D** IL-10 and **E** TNF. **F** Heatmaps displaying the normalised read counts of selected cytokines from RNA-seq analysis of monocytes treated with glucose or fructose (both 11.1 mM) stimulated with LPS (10 ng/mL) for 24 h. **G** Representative immunoblot and pooled data of downstream mTOR target pS6 with housekeeping control, actin. Statistical significance was assessed using a one-sample *t* test (**A**–**E**) or an unpaired, two-tailed *t* test (**G**). Data are representative of five (**A**, **B**, **D**, **E**), four (**C**), three (**F**) and six independent experiments (**G**) and are expressed as mean ± SEM. Source data are provided as a Source Data file.

Therefore, a potential explanation for the elevated ATP-linked respiration observed is that fructose-treated monocytes are supporting a higher level of lipogenesis. We observed no differences in phosphorylation of enzymes that catalyse the citrate-derived fatty acid synthesis steps; ATP citrate lyase (ACLY) or acetyl-CoA carboxylase (Supplementary Fig. 3H). However, fructose-cultured, LPS-stimulated monocytes have increased levels of the lipid mediator, prostaglandin E2 and greater sensitivity to the ACLY inhibitor, BMS303141, with regards to cytokine production (Supplementary Fig. 3I, J). Taken together, these data demonstrate that fructose treatment elevates ATP-linked oxygen consumption, independent of mitochondrial ROS and content.

**Fructose increases TCA cycling and anaplerosis.** To further characterise the metabolic activity of the mitochondria in glucose- and fructose-treated cells, we used SITA coupled with gas chromatography-mass spectrometry (GC-MS). Here, human monocytes were activated with LPS for 24 h and incubated with either $^{13}C_6$-glucose or $^{13}C_6$-fructose (Fig. 5A). Mass isotopologue distribution (MID) analysis of the TCA cycle intermediates— succinate, fumarate and malate—and amino acids—glutamate and aspartate—highlighted that there was increased cycling in the fructose-cultured monocytes compared to glucose. This was indicated by a reduced proportion of the unlabelled form of the metabolite (m + 0) and an increased proportion of the labelled form (predominantly represented by m + 2) (Fig. 5B, C and Supplementary Fig. 4A).

The TCA cycle relies on other metabolites to replenish it, a process termed anaplerosis. Glutamine-derived carbon enters the TCA cycle and also contributes to TCA-mediated amino acid biosynthesis, such as aspartate. To investigate whether fructose treatment increases

glutamine anaplerosis, we performed SITA with $^{13}C_5$-glutamine. Here, monocytes were incubated with $^{13}C_5$-glutamine in the presence of either glucose or fructose and activated with LPS (Fig. 5C). The ass isotopologue distribution analysis of the TCA cycle metabolites—succinate, fumarate and malate—demonstrated an increase in the percentage of $^{13}C$ into these intermediates in the presence of fructose (represented as m + 4) (Fig. 5D and Supplementary Fig. 4B). Fructose-treated monocytes are able to incorporate elevated amounts of glutamine-derived carbon to the TCA cycle intermediates and amino acids in comparison to glucose-treated monocytes (Supplementary Fig. 4C–G). These data demonstrate that fructose treatment increases the proportion of both sugar and glutamine carbon into the TCA cycle to support the observed increased rates of OXPHOS.

**Fructose-treated human monocytes are vulnerable to metabolic challenge.** Thus far, we have demonstrated that fructose treatment promotes a reduced glycolytic rate and enhanced OXPHOS. Glucose and fructose are both metabolised by the enzyme HK, producing metabolites glucose 6-phosphate or fructose-6-phosphate respectively[24]. Therefore, we sought to investigate the expression levels of the two predominant isoforms of HK: HKI and HKII. We observed no difference in expression levels of HKI; however, HKII levels were increased in fructose-treated, LPS-stimulated monocytes, in comparison to glucose (Fig. 6A, B). To further delineate the role of HK in monocyte function, we utilised the HK inhibitor, 2-DG. We treated activated monocytes in glucose or fructose for 24 h with 2-DG before measuring cytokine production. Following treatment with 2-DG, production of IL-1β, IL-6 and TNF was largely unaffected in glucose-treated cells, with the exception of IL-10. By contrast, fructose-treated

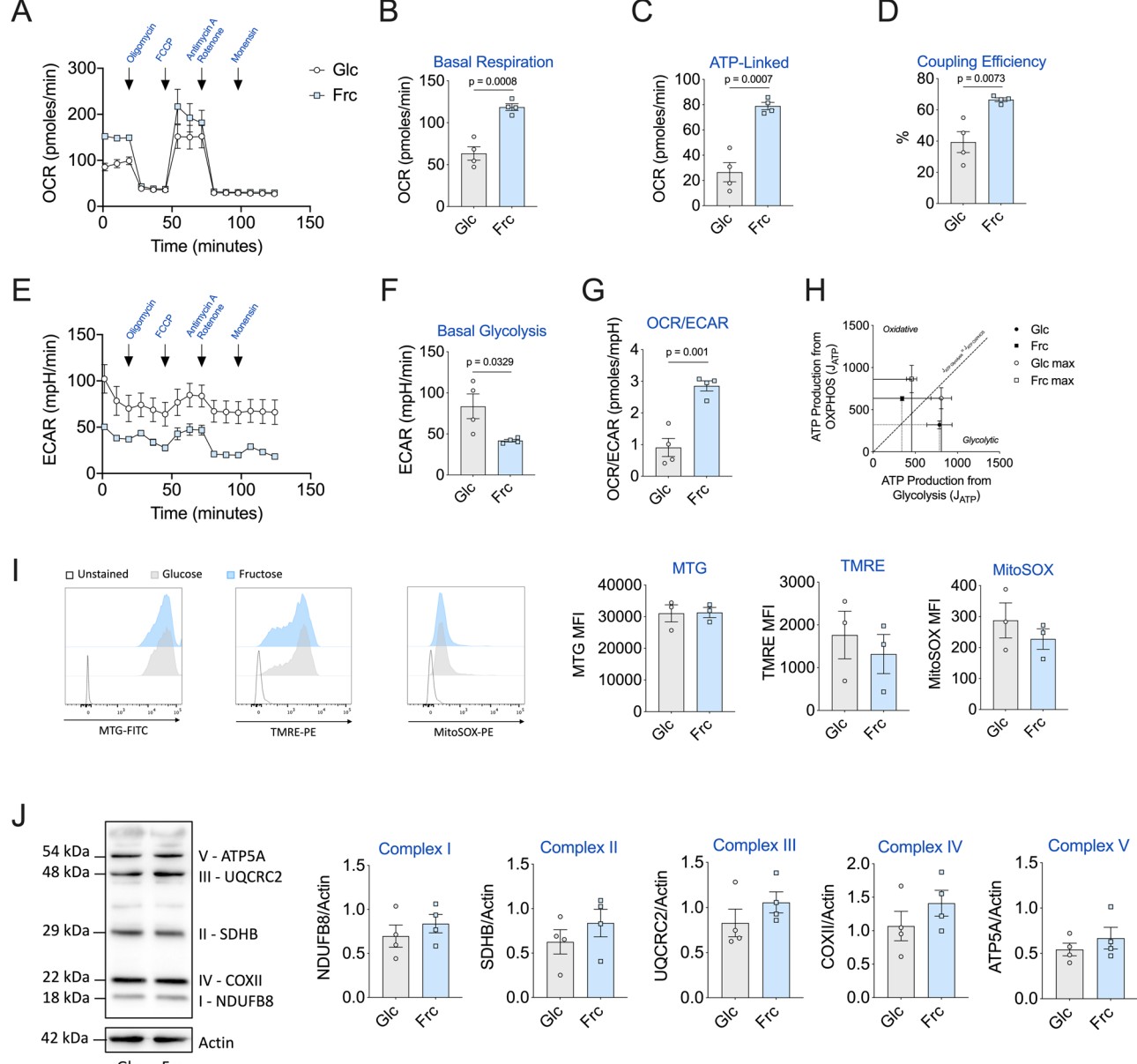

**Fig. 4 Fructose-induced oxidative phenotype is maintained in LPS-stimulated monocytes. A** Mitochondrial stress assay of monocytes cultured for 24 h in the presence of glucose or fructose (both 11.1 mM) and activated with LPS (10 ng/mL). Corresponding OCR measured with injections of oligomycin (1 µM), FCCP (3 µM), antimycin A (1 µM) and rotenone (1 µM) and monensin (20 µM). Respective mitochondrial parameters **B** basal respiration, **C** ATP-linked respiration and **D** coupling efficiency calculated via ATP-linked respiration/basal respiration. **E** Equivalent ECAR rate measured with **f** basal glycolysis levels. **G** OCR/ECAR ratio calculated by basal respiration/basal glycolysis levels. **H** Bioenergetic scope examining the ATP production ($J_{ATP}$) of oxidative phosphorylation versus glycolysis of basal and maximal of glucose or fructose. Mitochondrial parameters **i** Content: MitoTracker Green (MTG), membrane potential: tetramethylrhodamine ethyl ester (TMRE) and mitochondrial-derived ROS: MitoSOX measured by flow cytometry. **J** Mitochondrial respiratory complex (I–V) assessment by western blot. Statistical significance was assessed using an unpaired, two-tailed *t* test (**B–D**, **F–B**). Data are representative of four (**A–H**, **J**) or three (**I**) independent experiments. Data are expressed as mean ± SEM. Source data are provided as a Source Data file.

cells exhibited a dose-dependent decrease in all cytokines (Fig. 6C).

Having shown that 2-DG treatment reduced OXPHOS in fructose-treated cells (Fig. 1D), these results suggest the oxidative metabolic phenotype induced by fructose supports monocyte function. Given the enhanced reduction of cytokine expression, we next determined the level of cell viability for 2-DG-treated monocytes. Fructose-treated monocytes were acutely sensitive to 2-DG-mediated cell death in comparison to glucose-treated monocytes that were largely unaffected (Fig. 6D). We next wanted to assess whether the cells were similarly sensitive to inhibition of

oxidative metabolism. The viability of LPS-stimulated monocytes treated with glucose was minimally affected by inhibition of complex I (with rotenone), III (with antimycin A) and V (with oligomycin). However, monocytes treated with fructose were completely unable to tolerate incubation with these drugs, reflected by a striking reduction in viability (Fig. 6E). These data demonstrate that fructose treatment renders LPS-stimulated monocytes metabolically inflexible and dependent on oxidative metabolism. Sensitivity to both 2-DG and mitochondrial inhibitors suggest that glycolysis and oxidative metabolism become inextricably coupled upon exposure to fructose.

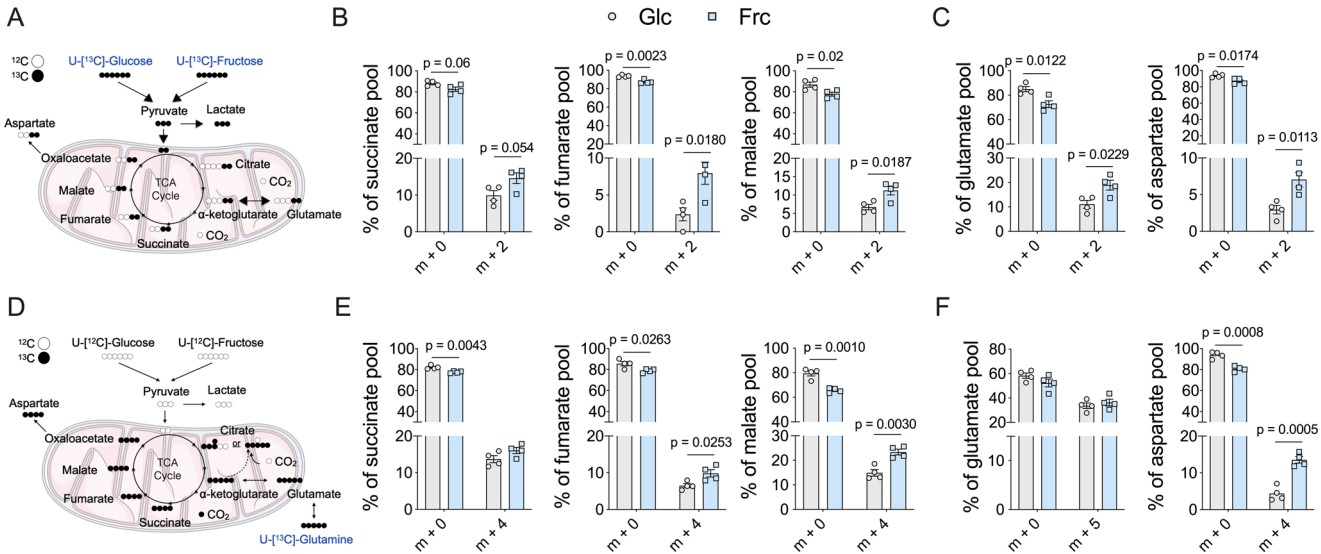

**Fig. 5 Fructose treatment induces elevated metabolic cycling. A** Stable isotope tracing of uniformly labelled $^{13}$C-glucose or $^{13}$C-fructose into the TCA cycle. **B** Mass isotopologue distribution (MID) represented as a % pool of TCA cycle metabolites: succinate, fumarate and malate or **C** amino acids: glutamate and aspartate of 24-h LPS-stimulated monocytes. Numbers on the *x*-axis represent the number of $^{13}$Carbons incorporated. **D** Stable isotope tracing of uniformly labelled $^{13}$C-glutamine in the presence of $^{12}$C-glucose or $^{12}$C-fructose of 24-h LPS-stimulated monocytes. **E** MID of succinate, fumarate and malate and **F** amino acids: glutamate and aspartate. Statistical significance of individual mass isotopomers was assessed using an unpaired, two-tailed *t* test (**B**, **C**, **E**, **F**). Data are representative of four independent experiments and are expressed as mean ± SEM. Source data are provided as a Source Data file.

**Dietary fructose increases inflammation in a mouse model.** Thus far, we have explored the metabolic and mechanistic implications of fructose exposure in human monocytes cultured ex vivo. In order to further investigate the impact of physiological fructose exposure on inflammation in vivo, we employed a mouse model. First, we determined whether mouse LPS-challenged bone marrow-derived macrophages (BMDMs) phenocopied human monocytes when exposed to fructose in vitro.

To better recapitulate the in vivo microenvironment, we incubated mouse macrophages with either glucose alone or a 1:1 ratio of glucose to fructose (maintaining the equivalent concentration of total monosaccharide). This strategy has been used previously in several studies to investigate the physiological impact of fructose exposure in an environment when glucose is also present[10,15,25]. Consistent with human monocytes, LPS-stimulated mouse macrophages exposed to fructose produced elevated levels of cytokines TNF, IL-1β, IL-6 and IL-12 at the protein level, but not the messenger RNA level in comparison to glucose alone (Fig. 7A and Supplementary Fig. 5A, B). Importantly, we demonstrated that this observation is not due to reduced glucose availability in the double monosaccharide condition (Supplementary Fig. 5C).

In order to confirm fructose uptake in the presence of glucose, we performed SITA analysis using mouse macrophages cultured in universally labelled $^{13}$C$_6$-glucose alone or $^{13}$C$_6$-glucose with $^{13}$C$_1$-fructose (Fig. 7B). Indeed, fructose enters the cells in the presence of glucose as depicted by the presence of the m + 1 isotopologue (Fig. 7C). Similar to human monocytes, BMDMs exposed to fructose significantly increased glutamine uptake and phosphorylation of the mTORC1 target, S6 ribosomal protein (Fig. 7D, E). Fructose treatment also led to an increase in Akt phosphorylation (Supplementary Fig. 5D).

Next, we investigated the role of glutamine metabolism in fructose-treated mouse macrophages and their response to LPS. The glutaminase inhibitor CB-839 did not alter levels of TNF; however, it significantly reduced IL-1β and IL-12 in BMDMs cultured in the presence of both monosaccharides, whereas cytokine production was unchanged in cells cultured with glucose

alone (Fig. 7F). CB-839 also reduced phosphorylation of S6 in fructose-exposed cells (Supplementary Fig. 5E), suggesting that increased glutamine metabolism supports mTORC1 activity in the presence of fructose. Further exploring the role of mTORC1 in fructose-mediated inflammation, we treated BMDMs with the mTORC1 inhibitor, rapamycin. Here, rapamycin treatment significantly reduced cytokine production in both glucose-alone and glucose–fructose-treated macrophages (Fig. 7G). These data suggest that mouse macrophages require mTORC1 activity for cytokine production regardless of sugar exposure, yet those exposed to both monosaccharides (in contrast to those exposed to glucose alone) rely on glutaminolysis to support the increased cytokine production promoted by fructose exposure.

Finally, to determine if fructose supplementation can influence LPS-induced inflammation in vivo, we used a mouse LPS model of systemic inflammation. Here, mice were provided 10% glucose or 10% fructose/10% glucose solutions for 2 weeks (to preclude development of any metabolic disorders) prior to LPS challenge (Fig. 7H). Strikingly, serum IL-1β levels were significantly increased in mice exposed to fructose, and an increasing trend of IL-6 and TNF was observed (Fig. 7I). This increase in serum IL-1β was not due to fructose-enhancing baseline IL-1β secretion, as seen at weeks 1 and 4 of sugar-water treatment in unchallenged mice (Supplementary Fig. 6A, B).

Interestingly, the presence of fructose does not result in a global increase in serum inflammatory markers, as while levels of CXCL5 were elevated in the presence of fructose, this did not reach statistical significance and there was no change in CXCL1 and CCL11 (Supplementary Fig. 6C–E). These data demonstrate that short-term fructose supplementation enhances LPS-induced systemic inflammation, suggesting physiological repercussions of high fructose exposure in mammals.

**Fructose-supported inflammation does not occur in T cells.** Given our results of an effect of fructose on both human and mouse mononuclear phagocytes, we were intrigued as to whether fructose exposure might have an effect on other immune cell types. However, functional observations of heightened

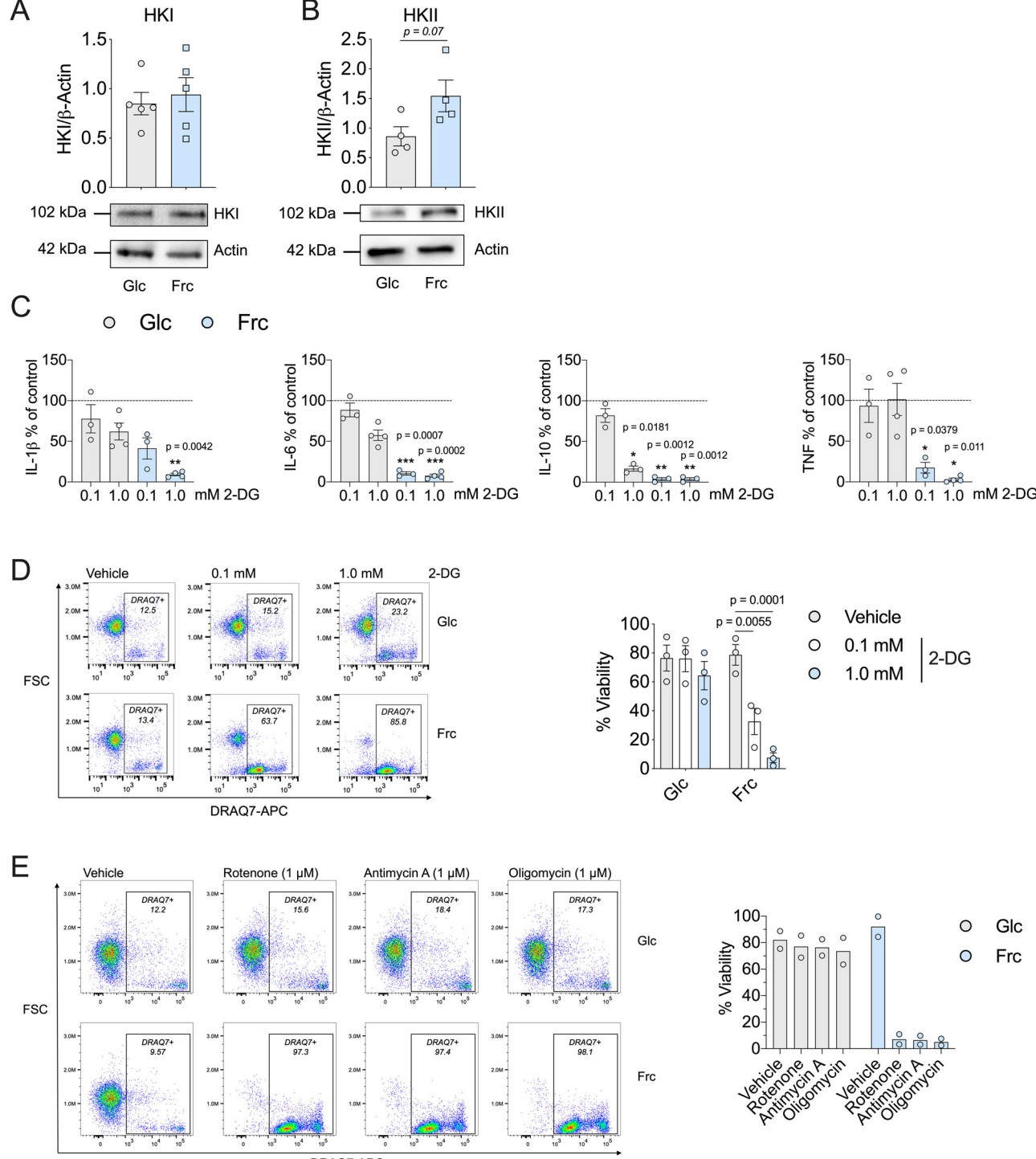

**Fig. 6 Fructose-induced inflammatory monocytes are sensitive to metabolic inhibition.** Immunoblot of LPS-stimulated monocytes in the presence of glucose or fructose with **A** hexokinase I or **B** hexokinase II analysed. **C** Cytokine release (% of control) of IL-1β, IL-6, IL-10 and TNF (dotted line indicates control) and **D** representative flow cytometry plot including DRAQ7 viability of glucose or fructose LPS-stimulated monocytes in the presence or absence of glycolytic inhibitor, 2-DG (0.1 or 1.0 mM). **E** Representative flow cytometry plot including DRAQ7 viability of glucose or fructose LPS-stimulated monocytes in the presence or absence of complex I inhibitor, rotenone (1 μM), complex III inhibitor antimycin A (1 μM) or complex V inhibitor, oligomycin (1 μM). Statistical significance was assessed using an unpaired, two-tailed $t$ test (**A**, **B**) or a two-way ANOVA with Sidak's multiple comparison test (**C**, **D**). Data are representative of five (**A**), four (**B**), three–four (**C**), three (**D**) or two independent experiments (**E**) and are expressed as mean ± SEM; *$p \leq 0.05$, **$p \leq 0.01$ and ***$p \leq 0.001$. Source data are provided as a Source Data file.

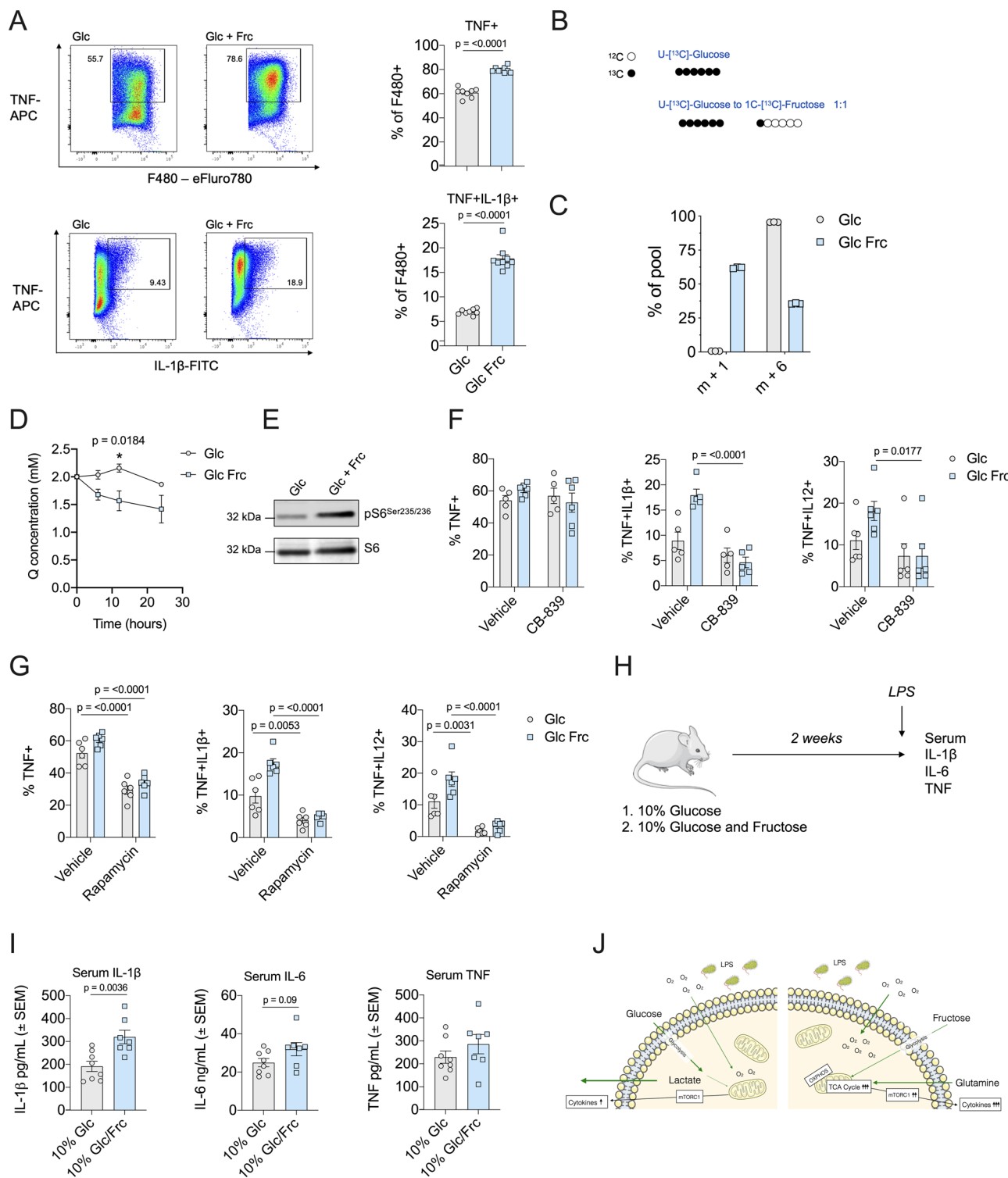

inflammation were not reflected in the mouse T cell compartment. We did not observe any differences in CD4+ or CD8+ T cell proliferation (Supplementary Fig. 7A). CD4+ T cells can be programmed into inflammatory subsets, such as T-helper type 1 (Th1) or anti-inflammatory subsets such as inducible regulatory T cells. Fructose supplementation did not influence polarisation of either Th1 or inducible regulatory T cells (Supplementary Fig. 7B, C). This suggests that fructose has a specific effect on LPS-stimulated mononuclear phagocytes.

## Discussion

Here, we investigate how activated human monocytes and mouse macrophages respond metabolically and functionally to fructose exposure. We demonstrate that mononuclear phagocytes from both species are metabolically plastic in engaging in the metabolism of an alternative carbon source and reprogramme cellular pathways to favour oxidative metabolism. Although able to rewire their metabolic pathways upon exposure to fructose, the cells are left metabolically inflexible and vulnerable to further metabolic

**Fig. 7 Fructose enhances inflammation in the presence of glucose in macrophages. A** Representative TNF and IL-1β flow cytometry plots and bar graphs of glucose (24 mM) or glucose and fructose (both 12 mM) cultured mouse macrophages treated with LPS (1 ng/mL) for 5 h in the presence of GolgiStop™. **B** Schematic of $^{13}C_6$-glucose (10 mM) or $^{13}C_6$-glucose (5 mM) and $^{13}C_1$-fructose (5 mM) isotope tracing. **C** Hexose isotopologues, m + 1 or m + 6, in mouse macrophages stimulated with LPS (1 ng/mL) for 24 h. **D** Glutamine uptake in the media of BMDMs cultured with 24 mM glucose or 12 mM glucose and 12 mM fructose stimulated with LPS (1 ng/mL) for 0, 12 and 24 h. **E** Immunoblot analysis for pS6$^{Ser235-236}$ in BMDMs stimulated with LPS overnight in the presence of glucose alone or glucose/fructose. Total S6 was used as a loading control. **F** Cytokine production assessed by flow cytometry and ICS for TNF, IL-1β and IL-12 of macrophages treated for 18 h with CB-839 (1 μM). **G** Cytokine production of TNF, IL-1β and IL-12 produced by macrophages cultured as **A** with or without rapamycin (50 μM). **H** Schematic of in vivo experiment. Mice fed a diet of 10% glucose (n = 8) or 10% glucose–fructose mixture (n = 7) for 2 weeks and stimulated with LPS (0.1 mg/kg) for 3 h (image of mouse obtained from Servier Medical Art). **I** Serum cytokine levels of IL-1β, IL-6 and TNF. **J** Schematic outlining fructose metabolism promoting inflammation. mTORC1 mammalian target of rapamycin complex 1, OXPHOS oxidative phosphorylation, TCA tricarboxylic acid. Statistical significance was assessed using an unpaired, two-tailed t test (**A**, **I**) or a two-way ANOVA with Sidak's multiple comparison test (**D**, **F**, **G**). Data are representative of five (**A**), two (**B**, **C**), three (**D–G**) or seven–eight independent experiments (**I**) and are expressed as mean ± SEM. Source data are provided as a Source Data file.

challenge. Importantly, we show that fructose exposure ex vivo promotes elevated cytokine production in both human and mouse mononuclear phagocytes and that a high fructose diet promotes an inflammatory phenotype in vivo, attributing pathophysiological relevance to our findings.

To date, most research has focused on the role of the disaccharide sucrose (glucose and fructose) in macrophage function. Sucrose stimulates pinocytosis in unstimulated macrophages and is broken down in the lysosome[26]. The only link to date between fructose and macrophage-mediated inflammation is that supraphysiological concentrations of sucrose (300 mM) enhance IL-1β secretion, most likely due to hypertonicity[27]. There is some evidence that fructose enhances myeloid function, either by increasing cytokine production in human dendritic cells or phagocytosis in mouse alveolar macrophages; however, the metabolic mechanisms of fructose-mediated inflammation have not been explored[17,28]. Coupled with this is a growing concern regarding high fructose corn syrup consumption globally—in some cases, accounting for 10% caloric intake in the USA[29]—thus underscoring the need to understand the implications of fructose exposure on macrophage function and inflammation.

Here, for the first time, we show that LPS-stimulated human monocytes and mouse macrophages exposed to fructose have an enhanced inflammatory phenotype supported by oxidative metabolism and glutaminolysis. We suggest this supports cytokine production through increased supply of biosynthetic intermediates. Mechanistically, we demonstrate that fructose-exposed cells have increased mTORC1 activity, and while this is required to support cytokine production regardless of sugar exposure, those cells exposed to fructose rely specifically on glutaminolysis to support their inflammatory phenotype (Fig. 7J). mTORC1 has previously been implicated in metabolic reprogramming in monocytes, with monocyte-specific deletion of Raptor (an mTORC1 scaffolding protein) leading to reduced oxidative metabolism during monocyte differentiation[30]. mTORC1 has also been shown to be activated by glutaminolysis and α-ketoglutarate production[31]. This is consistent with our findings and further suggests that mTORC1 can act both upstream and downstream of metabolic reprogramming.

We demonstrate that fructose-dependent metabolic reprogramming is maintained long term, but this is independent of changes to mitochondrial dynamics. Instead, it appears due to increased metabolic intermediate supply. Our data show that fructose carbon can contribute to both glycolysis and the TCA cycle, and whether this is mediated by HK or ketohexokinase in both human and mouse mononuclear phagocytes requires further exploration. In comparison to glucose, fructose-derived pyruvate is not converted to lactate. This could be due to reduced activity or expression of LDH (although our data would suggest otherwise), increased activity of pyruvate dehydrogenase and

mitochondrial pyruvate carrier 1 or decreased activity of pyruvate dehydrogenase kinase 1[32,33]. Fructose exposure has also been linked extensively to lipid biosynthesis, for example, fructose as a substrate is 30% more efficient at synthesising fatty acids than glucose, a phenomenon that has been implicated in the pathophysiology of non-alcoholic fatty liver disease[34,35]. Consistent with this, cytokine production was more sensitive to ACLY inhibition and levels of prostaglandin E2 were elevated in our fructose-treated monocytes.

It is clear that fructose levels fluctuate throughout health and disease. With the increased prevalence of high fructose diets in the Western world, understanding the impact of fructose on human health is critical. Fructose contributes to numerous metabolic disorders such as obesity, cancer and non-alcoholic fatty liver disease; however, to date, our understanding of its impact on the immune system is lacking[9,36,37]. A key strength of our study is the assessment of the impact of fructose on metabolism and inflammation across the two species. Akin to human monocytes, mouse macrophages increased consumption of glutamine, produced higher levels of cytokine production displayed and demonstrated elevated mTORC1 activity. A mouse model of LPS-induced inflammation allowed us to assess the physiological relevance of our findings. Previous in vivo studies of long-term high fructose exposure have been in the context of metabolic disorders such as steatosis and hyperglycaemia. Here, in order to circumvent any changes due to whole-body metabolism, we used a 2-week exposure strategy. This allowed us to demonstrate for the first time that fructose enhances inflammation independent of metabolic disease. Collectively, this provides direct evidence that fructose elevates inflammation under physiological conditions[38,39].

The increase in LPS-induced inflammation from dietary fructose was not due to an enhanced global inflammatory effect, with certain chemokines measured having no observable differences, in addition to no effect on the mouse T cell compartment. Whether this enhanced, fructose-mediated inflammation could contribute to downstream pathologies warrants further investigation. For instance, chronic fructose exposure and infection could heighten inflammation leading to non-alcoholic steatohepatitis or carcinogenesis[12]. In addition, our work using metabolic inhibitors shows that fructose treatment leaves cells metabolically inflexible and acutely vulnerable to further metabolic challenge. This highlights a potential vulnerability of human monocytes exposed to fructose when facing metabolically challenging environments, such as during bacterial infection (including sepsis) or in the tumour microenvironment, particularly in those individuals with a high fructose diet.

Our results have highlighted the metabolic plasticity of human monocytes in response to fructose exposure and have elucidated the metabolic mechanisms supporting fructose-induced inflammation. These findings highlight the importance of the microenvironment

in shaping the innate immune response and could form the foundations of investigations for therapies in areas as diverse as cancer and infectious diseases.

## Methods

**Human monocyte isolation and culture.** Peripheral blood was collected with informed written consent and ethical approval was obtained from Wales Research Ethics Committee 6 (13/WA/0190). Mononuclear cells were obtained by density gradient centrifugation (Histopaque-1077 (10771), Merck) Human CD14+ monocytes were isolated using CD14 microbeads (130-050-201; Miltenyi Biotec). The purity of monocytes was routinely monitored using anti-CD14 Pacific Blue (clone 63D3; 367122; BioLegend) via flow cytometry.

Monocytes ($1.0 \times 10^6$/mL) were activated with LPS (10 ng/mL; Ultrapure, tlrl-eblps; Invivogen) and cultured in glucose (G7021) or fructose (F3510; 11.1 mM; Merck) containing glucose-free RPMI (11879020; Thermo Fisher) supplemented with 10% dialysed foetal bovine serum (FBS; A3382001; Thermo Fisher Scientific). 2-DG (0.1–1 mM; D8375), oligomycin (1 μM; 75351), antimycin A (1 μM; A8674) and rotenone (1 μM; R8875) were obtained from Merck. The ACLY inhibitor, BMS303141 (4609), was purchased from Tocris. LPS purchased from Invivogen (*Escherichia coli* K12; tlrl-eklps) was used for the varying glucose concentration experiments.

Cells were harvested and analysed for flow cytometry (acquired with NovoExpress V1.4.1) and supernatants were stored at −20 °C for cytokine analysis.

**Mice.** Animal experiments were subject to ethical review by the Francis Crick Animal Welfare and Ethical Review Body and regulation by the UK Home Office project licence P319AE968. All mice were housed under conditions in line with the Home Office guidelines (UK). Mice were housed 3–5 per cage and were kept in a 12-h day/night cycle 07:00–19:00. Food and water were available ad libitum and rooms were kept at 21 °C at 55% humidity. All procedures were performed following the Animals (scientific procedures) Act 1986 and the EU Directive 2010.

C57/B6J mice were bred and housed at the Francis Crick Institute animal facility. All animals used were aged 6–15 weeks and littermates were randomly assigned to experimental groups.

**Mouse BMDM differentiation and culture.** Hind legs were collected from C57/B6J mice (aged 6–15 weeks) and cleaned using a scalpel, followed by flushing of the bone marrow with a 1 mL syringe, phosphate-buffered saline (PBS) and 25 G needle. Red blood cells were lysed using ACK lysis buffer following the manufacturer's instructions (10× red blood cell lysis buffer, 420301; BioLegend). Bone marrow cells were cultured on non-tissue culture-treated 10-cm Petri dishes in complete Iscove's modified Dulbecco's media (IMDM), 10% FBS, 50 μM β-mercaptoethanol and supplemented with 25 ng/mL of macrophage colony-stimulating factor (315-02; Peprotech). BMDMs were supplemented with macrophage colony-stimulating factor every 3 days until day 7 of differentiation. BMDMs were activated with 1 ng/mL of LPS purchased from Sigma (*E. coli* O111:B4 LPS25) for either 5 or 18 h according to experiment type. Further experiments were performed with the glutaminase inhibitor CB-839 (Merck, AMBH2D6FB23B) (1 μM per well) or rapamycin (R8781; Sigma) (50 nM per well) with their respective vehicle controls.

### Flow cytometry and intracellular cytokine staining

*Human monocytes.* Flow cytometry analysis was performed on monocytes after 24 h in culture. Cell death was monitored with DRAQ7 (1 μM, DR71000; BioStatus) and DRAQ7-negative cells were analysed with anti-HLA-DR (clone AC122; 130-095-293), anti-CD80 (clone REA661; 130-110-270), anti-CD86 (clone FM95; 130-113-572), anti-CCR5 (clone REA245; 130-117-356) and anti-CCR2 (clone REA624; 130-109-595; all Miltenyi Biotec) and anti-CD62L (clone DREG-56; 304806; BioLegend).

For mitochondria staining, cells were incubated with 100 nM MitoTracker Green (M7514; Thermo Fisher) for 30 min at 37 °C. For mitochondrial membrane potential and ROS, cells were incubated with tetramethylrhodamine ethyl ester 50 nM (ab113852; Abcam) and MitoSOX 5 μM (M36008; Thermo Fisher), respectively, for 20 min at 37 °C. Cells were acquired (Novocyte, ACEA) and downstream analysis was performed with FlowJo version 10 (TreeStar, USA).

*Mouse macrophages.* BMDMs were collected after 5 days of differentiation and switched into 24 mM glucose IMDM or 12 mM glucose/12 mM fructose IMDM for 2 days. BMDMs were plated in either in non-TC-treated 12- or 24-well plates ($1 \times 10^6$ cells and $0.5 \times 10^6$ BMDM, respectively). Intracellular cytokine staining for IL-12, TNF, IL-6 and pro-IL-1β (antibody details can be found below) was performed by adding 1 ng/mL of LPS and 0.8 μL of BD GolgiStop™ per 1 mL followed by 5 h of incubation at 37 °C.

BMDMs were scraped from the plastic and washed 1× in PBS. Surface staining was performed for F4/80 and viability dye (Viability Dye eFluor™ 780) was used to exclude dead cells. Cells were permeabilized, fixed, and stained with fluorescence-conjugated anti-IL-12, TNF, IL-6 or pro-IL-1β using the eBioscience™ Foxp3 Transcription Factor Staining Buffer Set.

*Antibodies.* Fluorescence-conjugated PerCP-Cy5.5 F4/80 (clone BM8; 123128), PE IL-6 (clone MP5-20F3; 504504) and PeCy7 IL-12/23 (clone C15.6; 505210) antibodies were purchased from BioLegend. FITC-IFNγ (clone XMG1.2; 35-7311), PE-FOXP3 (clone 3G3; 50-5773) and Violet Fluor450-CD4 (clone RM4-5; 750042) were purchased from Tonbo Biosciences. FITC-IL-1β-pro was purchased from Invitrogen (clone NJTEN3; 11-7114-82). TNF conjugated to APC (clone MP6-XT22; 506308) was purchased from eBioscience Life Technologies. All antibodies were used at a dilution of 1/300 for surface and intracellular staining.

Carboxyfluorescein succinimidyl ester stock at 5 mM was purchased from BioLegend (423801) and used at a final concentration of 85 nM. Viable cells were detected using the Fixable Viability dye eFluor780 (65-0865-14; eBioscience) at a dilution of 1/1000. Samples were acquired using BD FACSDiva V8.0.1 on the BD FACSymphony™ Flow Cytometer and data were analysed using FlowJo V10.3 (BD). An example gating strategy can be found in the Supplementary information (Supplementary Fig. 8).

**Metabolic analysis.** Metabolic analysis of monocytes was carried out using the Seahorse Extracellular Flux Analyzer (Agilent Technologies). Monocytes ($0.25 \times 10^6$ cells) were seeded onto a Cell-Tak (354240; Corning)-coated microplate allowing for immediate adhesion. To observe the glycolytic switch, cells were seeded in Seahorse XF assay media supplemented with 1% foetal calf serum (HyClone, 10703464; Fisher Scientific, USA) and 2 mM glutamine (G7515; Merck). Monocytes were given an initial injection of glucose, fructose, galactose (G5388; 11.1 mM; Merck) or media and allowed to equilibrate. LPS (10 ng/mL; Ultrapure, Invivogen) was injected after 1 h. To arrest glycolysis, a final injection of 2-DG (100 mM; Merck) was added. Corresponding OCR/ECAR (oxygen consumption rate/extracellular acidification rate) changes were monitored for the duration of the experiment. Alternatively, monocytes were given a third injection of oligomycin (1 μM) or GSK2837808A (LDHi, 10 μM; 5189; Tocris).

For the mitochondrial stress assay, monocytes were resuspended in Seahorse XF assay media supplemented with 11.1 mM glucose or fructose and 1 mM sodium pyruvate (103578-100; Agilent Technologies). Injections were of oligomycin (1 μM), FCCP (3 μM), antimycin A (1 μM), rotenone (1 μM) and monensin (20 μM; M5273; Sigma). Data were acquired using the Seahorse Wave software v2.6 (Agilent).

**Enzyme-linked immunosorbent assay.** IL-1β (DY201), IL-6 (DY206), IL-8 (DY208-05), IL-10 (DY217B), TNF (DY210) and prostaglandin E2 (KGE004B) were analysed using ELISA (DuoSets; R&D Systems).

Mouse serum samples were analysed using ELISA for IL-1β (Sigma, RAB0274), TNF (Mouse Uncoated ELISA Kit; Invitrogen, 88-7324-22), IL-6 (Mouse Uncoated ELISA Kit; Invitrogen, 88-7064-22), CXCL1 (Raybiotech, ELM-KC-1), CXCL5 (Raybiotech, ELM-LIX-1) and CCL11 (LEGEND MAX™, BioLegend, 443907).

ELISA plates were coated with the capture antibody and left overnight at 4 °C. Samples were appropriately diluted and incubated for 2 h at room temperature with gentle agitation, 2 h with the kit-specific secondary antibody and 20 min with streptavidin-horse radish peroxidase. The plate was then incubated at room temperature with a 1:1 mixture of hydrogen peroxide and tetramethylbenzidine (555214; BD Biosciences). Absorbance was measured at 450 nm after the addition of sulfuric acid (Merck) to each well and values were corrected to the blank.

**Extracellular lactate measurement.** Extracellular lactate was measured using the L-Lactate Assay Kit I (120001100A; Eton Bioscience). Samples and standards were appropriately diluted and mixed with the L-lactate assay solution and incubated at 37 °C for 30 min. Absorbance was then measured at 490 nm and concentrations calculated from the linear regression of the standard curve.

**RNA extraction and quantitative PCR of BMDMs.** Total RNA was extracted using RNeasy® columns (Qiagen) from three separate wells per condition according to the manufacturer's instructions. Genomic DNA was removed using On-Column DNA digestion (Qiagen). Complementary DNA (cDNA) was generated using the High-Capacity cDNA Reverse Transcription Kit (4368814; Thermo Fisher) according to the manufacturer's instructions. Power up™ SYBR® Green Master MIX (Applied Biosystem) was used to perform quantitative PCR. Primer sequences can be found in Supplementary Table 1. Gene expression values are calculated according to Pfaffl method[40] and are expressed as relative units compared to the control group. Quantitative PCR data were collected using Quant-Studio Design & Analysis software v1.3.

**RNA-seq sample preparation and sequencing.** Total RNA quality and quantity were assessed using Agilent 2100 Bioanalyser and an RNA Nano 6000 Kit (Agilent Technologies). Sequencing libraries were prepared with 100–900 ng of total RNA with an RNA integrity number value >8 using the Illumina® TruSeq® Stranded Total RNA with Ribo-Zero Gold™ Kit (Illumina Inc.). The steps included ribosomal RNA depletion and cleanup, RNA fragmentation, first-strand cDNA synthesis, second-strand cDNA synthesis, adenylation of 3′ ends, adapter ligation and PCR amplification (12 cycles). The manufacturer's instructions were followed except for the cleanup after the ribozero depletion step where Ampure®XP beads (Beckman Coulter) and 80% ethanol were used. Libraries were validated using the Agilent 2100 Bioanalyser

and a High-Sensitivity Kit (Agilent Technologies) to ascertain the insert size, and the Qubit® (Life Technologies) was used to perform the fluorometric quantitation. Following validation, the libraries were normalised to 4 nM, pooled together and clustered on the cBot™2 following the manufacturer's recommendations. The pool was then sequenced using a 75-base paired-end (2 × 75 bp PE) dual index read format on the Illumina® HiSeq4000 according to the manufacturer's instructions.

**RNA-seq analysis and differential gene expression.** Raw sequencing files were trimmed to remove adapter sequences and poor quality reads using TrimGalore. Trimmed reads were aligned with the STAR aligner (v2.5.1b) to the GRCh38 assembly of the human genome[41]. Raw counts were then calculated using Subread featureCounts (v1.5.1)[42], and a differential expression analysis comparing glucose- and fructose-treated monocytes was performed using DESeq2 (Bioconductor)[43]. The resultant p values were corrected for multiple testing using the Benjamini and Hochberg method[44]. Differentially regulated genes were defined as those with a log₂ FC > ±1 and an adjusted p value <0.05. Heatmaps were generated using Morpheus (Broad Institute).

**Immunoblot**

*Human monocytes.* Monocyte lysate proteins were quantified, denatured and separated using sodium dodecyl sulfate-polyacrylamide gel electrophoresis. Polyvinylidene difluoride membranes were probed with antibodies targeting HKI (2024), HKII (2867), phospho-S6 ribosomal protein (pS6; Ser235-236; 4858), phospho-AMPKa (2535), phospho-ACLY (4331), phospho-acetyl-CoA carboxylase 1 (3661), phospho-LDH (8176), pan-Akt (2920), phospho-AktSer473 (4060) and phospho-AktThr308 (4056). All antibodies were purchased from Cell Signaling (Danvers, MA) and used at a 1:1000 dilution. The total OXPHOS human cocktail (ab110411) and GLUT5 (ab41533) were purchased from Abcam and used at a dilution of 1:200 and 1:700, respectively. Protein loading was evaluated and normalised using β-actin (ab8226; Abcam). Densitometry on non-saturated immunoblots was measured using ImageJ software (Fiji)[45].

*Mouse BMDM.* Macrophages were lysed using RIPA buffer supplemented with 1% sodium dodecyl sulfate and phosphatase inhibitors (La Roche Ltd), denatured at 95 °C, and resolved on NuPAGE polyacrylamide pre-cast gels (Thermo Fisher Scientific) before the transfer of gels onto nitrocellulose membranes using the iBlot2 (Invitrogen). Lysates were probed for phospho-S6 Ser235/236 (4858) ribosomal protein, total rpS6 (clone 54D2; 2317), phospho-AktThr308 (clone 244F9; 4056), phospho-AktSer473 (clone D9E; 4060) and pan-Akt (clone 40D4; 2920). All antibodies were purchased from Cell Signaling Technologies. Secondary antibodies were purchased from LI-COR IRDye 800CW (donkey anti-mouse; 926-32212) and 680LT (donkey anti-rabbit; 926-68023). Fluorescence analysis was collected using the LI-COR Image Studio software v5.2. All original uncropped blots can be found in the Supplementary information (Supplementary Fig. 9).

**Gas chromatography-mass spectrometry.** Isolated monocytes were incubated with heavily labelled ¹³C₅-glutamine (2 mM; CLM-1822; Cambridge Isotopes) in RPMI phenol red-free media (Agilent Technologies) supplemented with 11.1 mM glucose or fructose and 10% dialysed FBS (Thermo Fisher Scientific). Alternatively, monocytes were cultured with ¹³C₆-glucose (CLM-1396) or ¹³C₆-fructose (CLM-1553; 11.1 mM; Cambridge Isotopes) in glucose-free RPMI (Thermo Fisher Scientific) supplemented with 10% dialysed FBS (Thermo Fisher Scientific). Monocytes were activated with LPS (10 ng/mL) for 24 h, washed twice with ice-cold saline and lysed in 80% methanol. Cell extracts were then dried down at 4 °C using a speed-vacuum concentrator.

Cellular metabolites were extracted and analysed by GC-MS using protocols described previously[46,47]. Metabolite extracts were derived using MTBSTFA (N-(tert-butyldimethylsilyl)-N-methyltrifluoroacetamide). D-Myristic acid (750 ng/sample) was added as an internal standard to metabolite extracts, and metabolite abundance was expressed relative to the internal standard. GC/MS analysis was performed using an Agilent 5975C GC/MS equipped with a DB-5MS + DG (30 m × 250 μm × 0.25 μm) capillary column (Agilent J&W, Santa Clara, CA, USA). Data were acquired using ChemStation E.02.02.1431. For SITA experiments, mass isotopomer distribution was determined using a custom algorithm developed at the McGill University[46].

**Liquid chromatography-mass spectrometry.** ¹³C₆-glucose was purchased from Cambridge Isotope Laboratories and D-fructose-¹³C₆ (587621) was purchased from Sigma-Aldrich. SITA with ¹³C-glucose and ¹³C-fructose allowed for the identification of isotopomer distribution of metabolites. Both ¹³C fructose and glucose were added in glucose-free Dulbecco's modified Eagle's medium supplemented with 10% dialysed FBS and 50 μM β-mercaptoethanol in the presence of LPS (10 ng/mL). Metabolites were extracted after 24 h of LPS activation and analysed by liquid chromatography-mass spectrometry (LC-MS) using methods previously described[48]. In brief, 1 × 10⁶ BMDMs were washed with cold PBS and metabolites were extracted with 200 μL of ice-cold extraction buffer (methanol, acetonitrile and water (50:30:20)). Samples were centrifuged at 21.1 × g for 10 min at 4 °C and the supernatant was collected for LC-MS analysis. Extracellular metabolites were extracted using 10 μL of culture media added to 490 μL of ice-cold extraction buffer

and centrifuged at the aforementioned speed. Supernatants were collected and run through LC-MS analysis. LC-MS machine information and operation is further described in Labuschagne et al.[49]. Spectra analysis was performed using the Thermo TraceFinder software.

**LPS-induced systemic inflammation model.** Female C57/B6J mice aged 8–10 weeks were randomly assigned either to 10% glucose water or 10% glucose/fructose (stock solution at 60% fructose and 40% glucose) water for 2 weeks prior to LPS challenge. Mice were injected intraperitoneally with LPS from E. coli (0111:B4; Sigma L4391) at a dose of 0.1 mg/kg. Mice were sacrificed 3 h post challenge and blood collected by cardiac puncture into EDTA-coated collection tubes. Blood was spun at 2000 × g for 15 min at 4 °C and serum was collected and stored at −80 °C.

**Data analysis.** Statistical analysis was performed using GraphPad Prism version 9 (USA). Data are represented as the mean ± standard error of the mean with significance taken as *p ≤ 0.05, **p ≤ 0.01 and ***p ≤ 0.001.

**Reporting summary.** Further information on research design is available in the Nature Research Reporting Summary linked to this article.

## Data availability

RNA-seq data have been deposited in GEO under the accession code GSE164058. All other data are available upon request or can be found within the manuscript and Supplementary information files. Source data are provided with this manuscript.

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

## Acknowledgements

We thank E. Ma, D. Elder and D. Skibinski for useful discussion, D. Avizonis and L. Choinière from McGill University Metabolomics Core Facility, the staff in the Joint Clinical Research Facility for phlebotomy and all blood donors. We also thank the Metabolomics Facility at the Francis Crick Institute for their support. We acknowledge the Wales Gene Park for their insight, expertise and technical support in generating the NGS data that assisted this research. Wales Gene Park is a Health and Care Research Wales funded infrastructure support group. This work was supported with grants awarded by Life Sciences Research Network Wales (NRN). G.W.J. is funded by a Versus Arthritis Career Development Fellowship (20305). J.B. was funded by a Canadian Institute for Health Research Postdoctoral Fellowship and the Kuok Family Postdoctoral Fellowship. E.E.V. is supported by a Diabetes UK RD Lawrence Fellowship (17/0005587) and by Cancer Research UK (C18281/A29019). This work was funded by Cancer Research UK Grants C596/A10419 and C596/A26855, and supported by the Francis Crick Institute, which receives its core funding from Cancer Research UK (FC001557), the UK Medical Research Council (FC001557) and the Wellcome Trust (FC001557).

## Author contributions

N.J., J.B., F.Z., A.R., B.J.J. and A.I.M.B. performed experiments. N.J., J.B., F.Z., C.J.B., D. M., J.G.C., D.K.F., K.H.V., E.E.V. and C.A.T. designed the experiments and provided intellectual discussion. N.J., J.B., F.Z., D.G.H., C.J.B., D.A., G.W.J. and E.E.V. analysed the data. N.J., J.B., D.K.F., E.E.V. and C.A.T. wrote the manuscript. All authors critically revised and approved the manuscript.

## Competing interests

K.H.V. is on the board of directors and shareholder of Bristol Myers Squibb, a shareholder of GRAIL, and on the science advisory board of PMV Pharma, RAZE Therapeutics, Volastra Pharmaceuticals and Ludwig Cancer. K.H.V. is a co-founder and consultant of Faeth Therapeutics, funded by Khosla Ventures. All other authors declare no competing interests.
