## [Peer Review File · Nature Communications]

REVIEWER COMMENTS

Reviewer #1 (Remarks to the Author):

Jones et al. report on the response of human macrophages to LPS when cultured in the presence of fructose as compared to glucose. They find different metabolic profiles linked to heightened cytokines responses in the presence of fructose.

The study has several positives as it is performed with human monocytes/macrophages while most studies are performed with murine macrophages and it is known how different they can respond to metabolic stress. Also they make use of state-of-the-art metabolic tracer analyses, Seahorse, etc.

However I am of the opinion that the manuscript requires a major review before it can be re-considered for publication, as I believe there are some important conceptual and technical aspects to be addressed:

1) There is no effective physiological or pathological context to the study. They discuss a lot of fructose as a nutrient restrictive condition but I believe this needs a major re-thinking. I understand that their idea of nutrient restriction is due to the fact that relative to glucose, fructose causes reduced glycolytic flux. But this is simply a cellular observation in an *in vitro* context. In physiology and pathology, as consumption of fructose has risen markedly in recent decades owing to the use of sucrose and high-fructose corn syrup in beverages and processed foods¹, this has contributed to increasing rates of obesity and non-alcoholic fatty liver disease (Wellen's group Nature 2020). They do not discuss nor test in which conditions, physiological or pathological macrophages have to deal with fructose. This makes their study an *in vitro* exercise so far, taking away much of its potential importance.

2) Related to 1, but more technical: they say that fructose levels are between 0.04mM and 0.2mM and can reach up to 5mM in specific conditions. Glucose levels in the blood are 5mM and 25mM is not uncommon in diabetics. Yet they carry out all their experiments comparing 11mM glucose to 11mM fructose. According to their own intro, cells in the body never see such concentrations of fructose. They need to cleverly rethink of all their experiments and perform the key ones at lower concentrations of fructose as compared to similar concentrations of glucose. Human macrophages are more resilient than murine to these metabolic perturbations, so they should have enough room to play.

3) Owing to the Wellen's paper in Nature 2020, which should be cited here, can they check whether macrophages in fructose are more likely to use lipid biosynthetic pathways? And can that explain some of their functional (cytokines) responses?

Reviewer #2 (Remarks to the Author):

The main aim of this study is to investigate the effect of carbon sources, mainly glucose or fructose, on cell metabolism and inflammatory response in primary human monocytes. The authors show that monocytes treated with fructose displayed low extracellular acidification rate (ECAR) and increased oxygen consumption rate (OCR), compared to monocytes treated with glucose. These data suggest that fructose reprograms cellular pathways in monocytes to favor oxidative metabolism. Despite the metabolic shift toward oxidative phosphorylation (OXPHOS), fructose-treated monocytes displayed increased expression of Hexokinase II (HKII), the first and rate-limiting enzyme in glycolysis. The authors further demonstrate that HKII-mediated glycolysis was tightly coupled to the tricarboxylic acid (TCA) cycle to support the increased rate of OXPHOS in fructose-treated monocytes. Fructose also promoted glutamine anaplerosis to further support the increased rates of OXPHOS. Furthermore, fructose-treated monocytes produced more inflammatory cytokines with no change in their transcripts. Since phosphorylation of ribosomal protein S6 was increased in monocytes treated with fructose compared to glucose, the authors concluded that increased mTORC1 activity and thereby translation is a mechanism for the

increased cytokine production in fructose-treated monocytes. Finally, monocytes treated with fructose were profoundly vulnerable to inhibition of glycolysis or OXPHOS.

The observations of how fructose reprograms cell metabolism and inflammatory responses in human monocytes are intriguing. However, the presented data are rather incomplete and lack underlying mechanistic understanding. The following suggestions would provide mechanistic insights on their observations and make the story complete.

Major points

1. The authors show that HKII inhibition by 2-deoxyglucose (2DG) caused a dose-dependent decrease in protein expression of inflammatory cytokines (Figure 5). Is the reduced cytokine production due to a decrease in mTORC1 activity? The authors should examine the effect of 2DG on S6 phosphorylation in fructose-treated monocytes.
2. The authors conclude that the increased translation is responsible for inflammatory cytokine production based solely on increased S6 phosphorylation in fructose-treated monocytes compared to glucose-treated monocytes. Although it is well established that mTORC1 promotes translation, increased S6 phosphorylation is an indirect readout for translation. The authors should directly measure protein translation of cytokines by using S35-methionine or similar tracer approaches. Furthermore, they should examine whether mTORC1 inhibition by rapamycin or other mTOR kinase inhibitors (Torin1, INK128..etc.) suppresses cytokine production in fructose-treated cells. These experiments would strengthen the authors' conclusion that fructose promotes cytokine production by activation of mTORC1 and thereby translation.
3. Is the activation of mTORC1 upstream or downstream of metabolic reprogramming toward OXPHOS in fructose-treated cells? The authors should measure ECAR and OCR upon mTORC1 inhibition in fructose-treated monocytes.
4. Related to the point 3, it has been shown that glutaminolysis, the conversion of glutamine to alpha-ketoglutarate, activates mTORC1 (PMID: 22749528). Thus, increased glutamine anaplerosis and thus up-regulated glutaminolysis may explain increased mTORC1 activity in fructose-treated monocytes. The authors should block glutaminolysis by 6-diazo-5-oxo-L-norleucine (DON) or Bis-2-(5-phenylacetamido-1,2,4-thiadiazol-2-yl)ethyl sulfide (BPTES), and examine mTORC1 activity and cytokine production in fructose-treated monocytes. These experiments may reveal the mechanism underlying fructose-mediated mTORC1 activation.
5. This reviewer disagrees with the authors' claim that fructose mimics nutrient starvation (abstract and main text) in human monocytes. They drew this conclusion based solely on the observation that ECAR and OCR in fructose-treated monocytes are similar to those in carbon source-starved monocytes. However, mTORC1 activity, which should be inhibited upon nutrient starvation, was increased in monocytes treated with fructose compared to glucose. Furthermore, fructose had no effect on AMPK, which should be increased upon carbon source starvation. Thus, fructose-treated monocytes are not starved for nutrients. I suggest to remove the claim unless the authors can provide further justification.
6. Figure 5. The authors show that fructose promotes the flow of sugar-derived pyruvate into the TCA cycle, rather than converting it to lactate, to support an increased rate of OXPHOS. Although this observation is intriguing, they did not provide nor even discuss an underlying mechanism. Could fructose downregulate lactate dehydrogenase (LDH) expression/activity and thus direct pyruvate towards the TCA cycle in human monocytes. The authors should analyze LDH expression or activity in fructose- or glucose-treated monocytes. Reduced LDH expression/activity could also explain why an LDH inhibitor had no impact on ECAR and OCR in fructose-treated monocytes (Figure 2).

Minor points

7. Providing a schematic model would be helpful to understand the findings.

8. The authors used supraphysiological concentration of fructose (11.1 mM) compared to ~0.2 mM in circulation and 5 mM in bone marrow. Why?

Please find below our detailed responses to all reviewers' comments and concerns marked in blue. *Italicised text* indicates text quoted from the manuscript and *underlined italicised text* is text added in the revised version.

Reviewer #1:

Jones et al. report on the response of human macrophages to LPS when cultured in the presence of fructose as compared to glucose. They find different metabolic profiles linked to heightened cytokines responses in the presence of fructose.

The study has several positives as it is performed with human monocytes/macrophages while most studies are performed with murine macrophages and it is known how different they can respond to metabolic stress. Also they make use of state-of-the-art metabolic tracer analyses, Seahorse, etc.

However I am of the opinion that the manuscript requires a major review before it can be re-considered for publication, as I believe there are some important conceptual and technical aspects to be addressed:

We thank the Reviewer for their extensive comments regarding our manuscript and are very pleased that they think our manuscript has several positives. We believe their comments have helped us substantiate our conclusions and strengthen our manuscript. Please see below a detailed point-by-point response to all comments:

1) There is no effective physiological or pathological context to the study. They discuss a lot of fructose as a nutrient restrictive condition but I believe this needs a major re-thinking. I understand that their idea of nutrient restriction is due to the fact that relative to glucose, fructose causes reduced glycolytic flux. But this is simply a cellular observation in an in vitro context. In physiology and pathology, as consumption of fructose has risen markedly in recent decades owing to the use of sucrose and high-fructose corn syrup in beverages and processed foods, this has contributed to increasing rates of obesity and non-alcoholic fatty liver disease (Wellen's group Nature 2020). They do not discuss nor test in which conditions, physiological or pathological macrophages have to deal with fructose. This makes their study an in vitro exercise so far, taking away much of its potential importance.

We agree wholeheartedly with the reviewer, the concept of fructose mimicking a nutrient restrictive condition was not appropriate for our study. We agree that the manuscript is much better focused on the physiological and pathological conditions in which monocytes are exposed to fructose. This was requested by both reviewers and we agree this provides a more appropriate context to the study.

We have reworked the abstract and introduction to frame the manuscript around the physiological and pathological conditions in which monocytes are exposed to fructose such as obesity, fructose-mediated non-alcoholic fatty liver disease and haematological malignancies with reference to the Wellen group 2020 Nature paper and others:

'Fructose intake has increased substantially throughout the Western world, largely attributed to elevated sucrose and high fructose corn syrup consumption⁷ and is thought to exacerbate various non-communicable conditions such as obesity, type 2 diabetes and non-alcoholic fatty liver disease⁷. Chronic fructose consumption in these conditions has recently been shown to drive hepatic fructolysis, where the expression of lipogenic genes is enhanced⁸⁻¹⁰.

Typically, physiological levels of fructose in the circulation range from 0.04 - 0.2 mM¹¹, however there are several pathophysiological scenarios in which levels of fructose are elevated. For example, peripheral blood levels can exceed 1 mM in patients with haematological malignancies such as acute myeloid leukaemia (AML) and acute lymphoblastic leukaemia (ALL)⁶. In addition, fructose concentrations in the bone marrow microenvironment of haematological cancer patients can reach up to 5 mM⁶. Alterations in the glucose to fructose ratio, particularly when glucose is scarce enables AML blasts to significantly enhance fructose uptake⁶. Localised murine tissue microenvironments, such as the liver, kidneys and jejunum also have elevated levels of fructose metabolism¹². Therefore, there are various pathophysiological scenarios and tissue microenvironments where monocytes will be exposed to either equimolar concentrations of fructose and glucose or concentrations of fructose exceeding that of glucose.

The impact of elevated fructose exposure on the immune system has not been investigated extensively. Chronic fructose exposure in rats results in a more inflammatory phenotype of bone marrow mononuclear cells¹⁶. Whilst there is some evidence that LPS-stimulated human

dendritic cells are able to produce enhanced levels of pro-inflammatory cytokines when cultured in fructose as opposed to glucose, the underlying metabolic rewiring that enables this pro-inflammatory phenotype has not been investigated¹⁵.

We have also now used the discussion to explore the pathological implications of our findings with regards to the physiological scenarios where myeloid cells are exposed to elevated levels of fructose:

'It is clear that fructose levels fluctuate throughout health and disease. With the increased prevalence of high fructose diets in the Western world, understanding the impact of fructose on human health is critical. Fructose contributes to numerous metabolic disorders such as obesity, cancer and non-alcoholic fatty liver disease; however, to date, our understanding of its impact on the immune system is lacking^{7,41,42}. A key strength of our study is the assessment of the impact of fructose on metabolism and inflammation across the two species. Akin to human monocytes, murine macrophages increased consumption of glutamine, produced higher levels of cytokine production displayed and demonstrated elevated mTORC1 activity. A murine model of LPS-induced inflammation allowed us to assess the physiological relevance of our findings. Previous in vivo studies of long-term high fructose exposure have been in the context of metabolic disorders such as steatosis and hyperglycemia. Here, in order to circumvent any changes due to whole body metabolism, we used a 2-week exposure strategy. This allowed us to demonstrate for the first time that fructose enhances inflammation independent of metabolic disease. Collectively, this provides direct evidence that fructose elevates inflammation under physiological conditions^{43,44}.

'The increase in LPS-induced inflammation from dietary fructose was not due to an enhanced global inflammatory effect, with certain chemokines measured having no observable differences, in addition to no effect on the murine T-cell compartment. Whether this enhanced, fructose-mediated inflammation could contribute to downstream pathologies warrants further investigation. For instance, chronic fructose exposure and infection could heighten inflammation leading to non-alcoholic steatohepatitis or carcinogenesis¹⁰. In addition, our work using metabolic inhibitors shows that fructose treatment leaves cells metabolically inflexible and acutely vulnerable to further metabolic challenge. This highlights a potential vulnerability of human monocytes exposed to fructose when facing metabolically challenging environments, such as during bacterial infection (including sepsis)

or in the tumour microenvironment, particularly in those individuals with a high fructose diet.'

2) Related to 1, but more technical: they say that fructose levels are between 0.04mM and 0.2mM and can reach up to 5mM in specific conditions. Glucose levels in the blood are 5mM and 25mM is not uncommon in diabetics. Yet they carry out all their experiments comparing 11mM glucose to 11mM fructose. According to their own intro, cells in the body never see such concentrations of fructose. They need to cleverly rethink of all their experiments and perform the key ones at lower concentrations of fructose as compared to similar concentrations of glucose.

The reviewer raises an important issue, we are keen to make our findings as physiologically relevant as possible and move the study beyond an *in vitro* exercise. We have taken two strategies to address this in the revised version of the manuscript. Firstly, the revised *ex vivo* experiments were performed using a 1:1 ratio of glucose to fructose, this strategy has been used previously by several studies to investigate the physiological impact of fructose exposure in an environment when glucose is also present. Secondly, we conducted an *in vivo* experiment to assess the impact of a high fructose diet on systemic inflammation.

Due to the impact of COVID-19 we have been and remain unable to obtain human peripheral blood. However, through a fortuitous collaboration with the Vousden lab at The Francis Crick Institute we have been able to address the reviewers' comments using a murine model. We confirmed that mouse macrophages phenocopy the human cells when exposed to fructose and are therefore confident this is an appropriate model for further exploration. We believe this significantly strengthens the manuscript, allowing us to comment on the implications of fructose exposure in both species. Importantly, it has also allowed us to explore the consequences of fructose exposure *in vivo*.

After careful consideration we decided that the best representative *in vitro* and *in vivo* condition to reflect the physiological environment is the comparison of a 1:1 ratio of glucose and fructose to glucose alone. As aforementioned there are certain pathologies where fructose concentrations can match or exceed that of glucose, such as patients with various haematological cancers *Jang et al., 2018 Cell Metabolism* and *Jang et al., 2020 Nature Metabolism*. Fructose will invariably be present in the environment in addition to glucose,

therefore we decided it would be best to conduct our fructose exposure experiments in the presence of glucose and compare the results to that of glucose exposure alone. We therefore compare equimolar concentrations of glucose alone to a 1:1 combination of glucose and fructose both *in vitro* and *in vivo*. Importantly we have performed control experiments to ensure the phenotype observed upon fructose exposure is not simply due to reduced levels of glucose (Supplementary Figure 5C).

We show that in comparison to glucose alone, the addition of fructose increases cytokine production, mTOR activity and glutamine uptake in murine macrophages which are all key phenotypes that we observe in human monocytes. In addition, we confirm that fructose is taken up when glucose is present, further suggesting our model is physiologically relevant. We also now include intriguing data suggesting that the fructose exposure phenotype is specific to mononuclear phagocytes (we see no such effects of fructose on various murine T-cell compartments).

Importantly, we demonstrate *in vivo* that the presence of fructose in the diet (in combination with glucose) increases serum cytokine levels in comparison to glucose alone – providing evidence for a physiological effect of fructose on systemic inflammation. Use of a two-week long diet circumvented any effect from the mice developing a metabolic disorder and confounding the experiment. These data highlight there may be important pathological effects of a high fructose diet for mononuclear phagocyte function and we believe this addition to the manuscript provides the physiological context and relevance needed to elevate it beyond an *in vitro* exercise.

The manuscript has been extensively reworked and reemphasised to reflect the new data and focus on the physiological and pathological conditions in which monocytes are exposed to fructose.

3) Owing to the Wellen's paper in Nature 2020, which should be cited here, can they check whether macrophages in fructose are more likely to use lipid biosynthetic pathways? And can that explain some of their functional (cytokines) responses?

We thank the Reviewer for this important suggestion. Given several recent studies linking fructose exposure to lipogenesis it is important we explore this here. We have now included

data, obtained prior to lockdown, investigating the role of lipid biosynthetic pathways in human monocytes. We demonstrate that cytokine production in fructose treated monocytes was found to be more sensitive to the ATP citrate lyase inhibitor; BMS303141, suggesting they are more dependent on the lipid biosynthesis pathway. Secondly, we assessed levels of the lipid mediator prostaglandin E2 and show they are increased by fructose treatment in comparison to glucose treated monocytes. We have now added this data to Supplementary Figure 3 (H-J) and added the following text to the results and discussion sections of the manuscript:

'A high fructose diet has been shown to increase de novo lipogenesis in the liver⁸. This correlates with increased mitochondrial ATP production, which may support this energy demanding process^{27,28}. Therefore, a potential explanation for the elevated ATP-linked respiration observed is that fructose-treated monocytes are supporting a higher level of lipogenesis. We observed no differences in phosphorylation of enzymes that catalyse the citrate-derived fatty acid synthesis steps; ATP citrate lyase (ACLY) or acetyl-CoA carboxylase (ACC; Supplementary Figure 3H). However, fructose cultured LPS-stimulated monocytes have an increase in levels of the lipid mediator, prostaglandin E2 and greater sensitivity to the ACLY inhibitor BMS303141 with regards to cytokine production (Supplementary Figure 3I-J).'

Discussion:

'Fructose exposure has also been linked extensively to lipid biosynthesis, for example fructose, as a substrate, is 30% more efficient at synthesising fatty acids than glucose, a phenomenon that has been implicated in the pathophysiology of non-alcoholic fatty liver disease^{39,40}. Consistent with this, cytokine production was more sensitive to ACLY inhibition and levels of PGE2 were elevated in our fructose-treated monocytes.'

Reviewer #2:

The main aim of this study is to investigate the effect of carbon sources, mainly glucose or fructose, on cell metabolism and inflammatory response in primary human monocytes. The authors show that monocytes treated with fructose displayed low extracellular acidification rate (ECAR) and increased oxygen consumption rate (OCR), compared to monocytes treated with glucose. These data suggest that fructose reprograms cellular pathways in monocytes to

favor oxidative metabolism. Despite the metabolic shift toward oxidative phosphorylation (OXPHOS), fructose-treated monocytes displayed increased expression of Hexokinase II (HKII), the first and rate-limiting enzyme in glycolysis. The authors further demonstrate that HKII-mediated glycolysis was tightly coupled to the tricarboxylic acid (TCA) cycle to support the increased rate of OXPHOS in fructose-treated monocytes. Fructose also promoted glutamine anaplerosis to further support the increased rates of OXPHOS. Furthermore, fructose-treated monocytes produced more inflammatory cytokines with no change in their transcripts. Since phosphorylation of ribosomal protein S6 was increased in monocytes treated with fructose compared to glucose, the authors concluded that increased mTORC1 activity and thereby translation is a mechanism for the increased cytokine production in fructose-treated monocytes. Finally, monocytes treated with fructose were profoundly vulnerable to inhibition of glycolysis or OXPHOS.

The observations of how fructose reprograms cell metabolism and inflammatory responses in human monocytes are intriguing. However, the presented data are rather incomplete and lack underlying mechanistic understanding. The following suggestions would provide mechanistic insights on their observations and make the story complete.

We would like to thank the reviewer for their insightful comments on the manuscript and feel it is much improved following our revisions. Please see below detailed responses to all the comments.

As previously mentioned, due to the impact of COVID-19 we have been and remain unable to obtain human peripheral blood. However, through a fortuitous collaboration with the Vousden lab at The Francis Crick Institute we have been able to address the Reviewers' comments using a murine model. We have confirmed the murine macrophages phenocopy the human cells when exposed to fructose and have repeated our key experiments (Figure 7A-F) using a physiologically relevant condition (a 1:1 ratio of glucose to fructose). Experiments conducted to address reviewer 2's comments have therefore been conducted using these conditions.

Major points

1. The authors show that HKII inhibition by 2-deoxyglucose (2DG) caused a dose-dependent decrease in protein expression of inflammatory cytokines (Figure 5). Is the reduced cytokine

production due to a decrease in mTORC1 activity? The authors should examine the effect of 2DG on S6 phosphorylation in fructose-treated monocytes.

We would like to thank the Reviewer for their experimental suggestions regarding the role of mTORC1 in the mechanistic control of the metabolic reprogramming induced by fructose treatment and we have addressed this in response to their second major point below. Specifically, regarding the effect of 2DG on mTORC1, we have investigated this and include the result below. The experiment shows that pS6 levels were reduced in both glucose and glucose + fructose treated murine macrophages. The reason we have decided not to include these data in the manuscript is because 2DG causes a dramatic decrease in cell viability in fructose treated cells. This makes it difficult to attribute signalling changes to the specific effects of hexokinase inhibition rather than due to cell death.

Reviewer-only Figure 1: Representative immunoblot of pS6^{Ser235/236} of glucose (24 mM) or glucose + fructose (both 12 mM) cultured murine macrophages treated with LPS (1 ng/mL) for 18 hours in the presence or absence of 2-DG (2 mM).

2. The authors conclude that the increased translation is responsible for inflammatory cytokine production based solely on increased S6 phosphorylation in fructose-treated monocytes compared to glucose-treated monocytes. Although it is well established that mTORC1 promotes translation, increased S6 phosphorylation is an indirect readout for translation. The authors should directly measure protein translation of cytokines by using S35-methionine or similar tracer approaches. Furthermore, they should examine whether mTORC1 inhibition by rapamycin or other mTOR kinase inhibitors (Torin1, INK128..etc.) suppresses cytokine production in fructose-treated cells. These experiments would strengthen the authors' conclusion that fructose promotes cytokine production by activation of mTORC1 and thereby translation.

The reviewer raises an important point and we agree that investigation of the role of mTORC1 in the increased cytokine production upon fructose treatment was needed. To address this, we have performed experiments to assess cytokine production upon fructose exposure in the absence and presence of the mTORC1 inhibitor, rapamycin (now included in Figure 7G). Here, we employed a direct measure of protein translation (intracellular cytokine staining) that was achieved using a 5 hour time point, allowing the cells to translate available mRNA (Chang et al., 2013 and Jung et al., 1993). We demonstrate that rapamycin reduces cytokine production in both the glucose only and glucose + fructose treated murine macrophages. These results suggest that mTORC1 governs cytokine production in murine macrophages. Although mTORC1 activity is universally required for cytokine production in mononuclear phagocytes, following suggested experiments by this reviewer (see below) we have shown that glutaminolysis is specifically required for the elevated levels of cytokines in fructose-treated cells.

We have added the following text to the results and discussion sections of the manuscript to incorporate the new data:

Results:

'Further exploring the role of mTORC1 in fructose-mediated inflammation, we treated BMDMs with the mTORC1 inhibitor, rapamycin. Here, rapamycin treatment significantly reduced cytokine production in both glucose alone and glucose-fructose treated macrophages (Figure 7G). These data suggest that murine macrophages require mTORC1 activity for cytokine production regardless of sugar exposure, yet those exposed to both monosaccharides (in contrast to those exposed to glucose alone) rely on glutaminolysis to support the increased cytokine production promoted by fructose exposure.'

Discussion:

'Here, for the first time, we show that LPS-stimulated human monocytes and murine macrophages exposed to fructose have an enhanced inflammatory phenotype supported by oxidative metabolism and glutaminolysis. We suggest this supports cytokine production through increased supply of biosynthetic intermediates. Mechanistically we demonstrate that fructose exposed cells have increased mTORC1 activity and while this is required to support cytokine production regardless of sugar exposure, those cells exposed to fructose rely specifically on glutaminolysis to support their inflammatory phenotype (Figure 7J).'

3. Is the activation of mTORC1 upstream or downstream of metabolic reprogramming toward OXPHOS in fructose-treated cells? The authors should measure ECAR and OCR upon mTORC1 inhibition in fructose-treated monocytes.

The reviewer raises an important point, however, due to logistical restrictions imposed by COVID-19 we have not been able to conduct this experiment. However, the activation of mTORC1 and its relative position upstream or downstream of metabolic reprogramming has already been investigated extensively in the field. For example, a study by Lee et al., 2019 demonstrated that human monocytes, cultured with rapamycin + LPS resulted in a decreased glycolytic rate in comparison to LPS alone in the immediate term. This suggests an upstream role for mTORC1 in metabolic reprogramming in monocytes (Lee et al., 2019). In addition, during monocyte differentiation, Karmaus et al., 2017 revealed that Raptor deletion in myeloid cells resulted in a reduced oxidative metabolism (Karmaus et al., 2017). While this experiment is not in the context of fructose treatment, it does suggest that activation of mTORC1 is upstream of metabolic reprogramming resulting in increased OXPHOS.

We have added a section in the discussion to speculate on the likely position of mTORC1 in the sequence of events given the current literature in the field. This is also reflected in the new graphical Figure 7J.

‘Mechanistically we demonstrate that fructose exposed cells have increased mTORC1 activity and while this is required to support cytokine production regardless of sugar exposure, those cells exposed to fructose rely specifically on glutaminolysis to support their inflammatory phenotype (Figure 7J). mTORC1 has previously been implicated in metabolic reprogramming in monocytes, with monocyte-specific deletion of Raptor (an mTORC1 scaffolding protein) leading to reduced oxidative metabolism during monocyte differentiation³⁵. mTORC1 has also been shown to be activated by glutaminolysis and α -ketoglutarate production³⁶. This is consistent with our findings and further suggests the mTORC1 can act both upstream and downstream of metabolic reprogramming.’

In addition, the Sabatini group have recently reported that fructose can activate mTORC1 and that mTORC1 senses glycolytic activity via dihydroxyacetone phosphate (DHAP) (Orozco et al 2020). To acknowledge this we have added the following sentence to the results section:

‘Fructose has recently been reported to activate mTORC1 via dihydroxyacetone phosphate (DHAP) sensing²⁵. Consistent with this, phosphorylation of the downstream mTOR target, S6 ribosomal protein, was elevated significantly in LPS-stimulated monocytes treated with fructose (Figure 3G).’

4. Related to point 3, it has been shown that glutaminolysis, the conversion of glutamine to alpha-ketoglutarate, activates mTORC1 (PMID: 22749528). Thus, increased glutamine anaplerosis and thus up-regulated glutaminolysis may explain increased mTORC1 activity in fructose-treated monocytes. The authors should block glutaminolysis by 6-diazo-5-oxo-L-norleucine (DON) or Bis-2-(5-phenylacetamido-1,2,4-thiadiazol-2-yl)ethyl sulfide (BPTES), and examine mTORC1 activity and cytokine production in fructose-treated monocytes. These experiments may reveal the mechanism underlying fructose-mediated mTORC1 activation.

The reviewer raises an important point. To address this, we first confirmed that murine macrophages phenocopied the human monocytes by increasing their glutamine uptake in response to fructose (Figure 7D). Next, we investigated cytokine production and mTORC1 activity upon treatment with the glutaminase inhibitor; CB-839 in murine macrophages cultured either in glucose or a 1:1 ratio of glucose + fructose. We demonstrate that upon CB-839 treatment, macrophages treated with glucose + fructose have a significant reduction in their IL-1 β and IL-12 production and marginally decreased TNF α production in comparison to those cultured with glucose alone. mTORC1 activity was also reduced upon CB-839 treatment in fructose treated cells. To incorporate these data we have added the following to the manuscript:

Results:

‘Next, we investigated the role of glutamine metabolism in fructose-treated murine macrophages and their response to LPS. The glutaminase inhibitor CB-839, did not alter levels of TNF α ; however, it significantly reduced IL-1 β and IL-12 in BMDMs cultured in the presence of both monosaccharides, whereas cytokine production was unchanged in cells cultured with glucose alone (Figure 7F). CB-839 also reduced phosphorylation of S6 in fructose-exposed cells (Supplementary Figure 5E), suggesting increased glutamine metabolism supports mTORC1 activity in the presence of fructose.’

Discussion:

'Here, for the first time, we show that LPS-stimulated human monocytes and murine macrophages exposed to fructose have an enhanced inflammatory phenotype supported by oxidative metabolism and glutaminolysis. We suggest this supports cytokine production through increased supply of biosynthetic intermediates. Mechanistically we demonstrate that fructose exposed cells have increased mTORC1 activity and while this is required to support cytokine production regardless of sugar exposure, those cells exposed to fructose rely specifically on glutaminolysis to support their inflammatory phenotype (Figure 7J).'

5. This reviewer disagrees with the authors' claim that fructose mimics nutrient starvation (abstract and main text) in human monocytes. They drew this conclusion based solely on the observation that ECAR and OCR in fructose-treated monocytes are similar to those in carbon source-starved monocytes. However, mTORC1 activity, which should be inhibited upon nutrient starvation, was increased in monocytes treated with fructose compared to glucose. Furthermore, fructose had no effect on AMPK, which should be increased upon carbon source starvation. Thus, fructose-treated monocytes are not starved for nutrients. I suggest to remove the claim unless the authors can provide further justification.

We agree with the reviewer that our claims regarding fructose mimicking nutrient restriction need revising. Our manuscript is much better focused on the physiological and pathological conditions in which monocytes are exposed to fructose. This was requested by both reviewers and we agree this provides a more appropriate context to the study.

We have reworked the abstract and introduction to frame the manuscript around the physiological and pathological conditions in which monocytes are exposed to fructose such as obesity, fructose-mediated non-alcoholic fatty liver disease and haematological malignancies.

In addition, we have removed all claims in the revised manuscript that fructose mimics nutrient restriction and focussed on the impact of fructose on human monocytes.

Introduction:

'Fructose intake has increased substantially throughout the Western world, largely attributed to elevated sucrose and high fructose corn syrup consumption⁷ and is thought to exacerbate various non-communicable conditions such as obesity, type 2 diabetes and non-alcoholic

fatty liver disease⁷. Chronic fructose consumption in these conditions has recently been shown to drive hepatic fructolysis, where the expression of lipogenic genes is enhanced⁸⁻¹⁰.

Typically, physiological levels of fructose in the circulation range from 0.04 - 0.2 mM¹¹, however there are several pathophysiological scenarios in which levels of fructose are elevated. For example, peripheral blood levels can exceed 1 mM in patients with haematological malignancies such as acute myeloid leukaemia (AML) and acute lymphoblastic leukaemia (ALL)⁶. In addition, fructose concentrations in the bone marrow microenvironment of haematological cancer patients can reach up to 5 mM⁶. Alterations in the glucose to fructose ratio, particularly when glucose is scarce enables AML blasts to significantly enhance fructose uptake⁶. Localised murine tissue microenvironments, such as the liver, kidneys and jejunum also have elevated levels of fructose metabolism¹². Therefore, there are various pathophysiological scenarios and tissue microenvironments where monocytes will be exposed to either equimolar concentrations of fructose and glucose or concentrations of fructose exceeding that of glucose.

The impact of elevated fructose exposure on the immune system has not been investigated extensively. Chronic fructose exposure in rats results in a more inflammatory phenotype of bone marrow mononuclear cells¹⁶. Whilst there is some evidence that LPS-stimulated human dendritic cells are able to produce enhanced levels of pro-inflammatory cytokines when cultured in fructose as opposed to glucose, the underlying metabolic rewiring that enables this pro-inflammatory phenotype has not been investigated¹⁵.

6. Figure 5. The authors show that fructose promotes the flow of sugar-derived pyruvate into the TCA cycle, rather than converting it to lactate, to support an increased rate of OXPHOS. Although this observation is intriguing, they did not provide nor even discuss an underlying mechanism. Could fructose downregulate lactate dehydrogenase (LDH) expression/activity and thus direct pyruvate towards the TCA cycle in human monocytes. The authors should analyze LDH expression or activity in fructose- or glucose-treated monocytes. Reduced LDH expression/activity could also explain why an LDH inhibitor had no impact on ECAR and OCR in fructose-treated monocytes (Figure 2).

We agree with the reviewer that fructose could indeed downregulate lactate dehydrogenase (LDH) activity. We have data from human monocytes (obtained prior to lockdown) that

fructose treatment does not affect the phosphorylation of LDH in comparison to glucose. Here, we performed a time course assessing the level of phospho-LDH across three time points, 0.25, 1 and 6 hours between glucose and fructose treated, LPS-stimulated monocytes. We have now added the data to Supplementary Figure 1G and the following text to the manuscript:

‘Secondly, to establish whether the elevated ECAR levels post-LPS treatment reflected glycolytic activity as opposed to other acidifying processes, we used a lactate dehydrogenase inhibitor (GSK2837808A; LDHi). Here, the increased ECAR upon LPS-stimulation was reduced in glucose-treated monocytes upon LDHi treatment (Figure 2D). By contrast, LDHi barely impacted ECAR in fructose-treated cells, arguing that fructose-mediated glycolysis is coupled to OXPHOS. The low level of ECAR under this condition is most likely due to acidification of the media by an alternative source to lactate (Figure 2D). We confirmed this was not due to changes in LDH phosphorylation in fructose-treated versus glucose-treated cells (Supplementary Figure 1F).’

In addition, we demonstrate that there is no change in LDHA expression between glucose or glucose and fructose treated murine macrophages.

Reviewer-only Figure 2: Representative immunoblot of LDHA expression of glucose (24 mM) or glucose and fructose (both 12 mM) cultured murine macrophages treated with LPS (1 ng/mL) for 18 hours. β -actin used as the loading control.

We have now speculated in the discussion regarding the mechanism for pyruvate import into the mitochondria as opposed to conversion to lactate:

‘However, in comparison to glucose, fructose-derived pyruvate is not converted to lactate. Mechanistically, this could be due to reduced activity or expression of LDH (although our data would suggest otherwise), increased activity of pyruvate dehydrogenase and mitochondrial pyruvate carrier 1 or decreased activity of pyruvate dehydrogenase kinase

1^{37,38}. Fructose exposure has also been linked extensively to lipid biosynthesis, for example fructose, as a substrate, is 30% more efficient at synthesising fatty acids than glucose, a phenomenon that has been implicated in the pathophysiology of non-alcoholic fatty liver disease^{39,40}. Consistent with this, cytokine production was more sensitive to ACLY inhibition and levels of PGE2 were elevated in our fructose-treated monocytes.'

Minor points

7. Providing a schematic model would be helpful to understand the findings. We have now added a schematic model to the manuscript illustrating our findings (Figure 7J).

8. The authors used supraphysiological concentration of fructose (11.1 mM) compared to ~0.2 mM in circulation and 5 mM in bone marrow. Why?

In addressing reviewer 1's comments we have taken steps to conduct revised experiments using a more physiological relevant condition. After careful consideration we decided that the best representative *in vitro* and *in vivo* condition to reflect the physiological environment is the comparison of a 1:1 ratio of glucose and fructose to glucose alone. Fructose will invariably be present in the environment in addition to glucose, therefore we decided it would be best to conduct our fructose exposure experiments in the presence of glucose and compare the results to that of glucose exposure alone. We therefore compare equimolar concentrations of glucose alone to a 1:1 combination of glucose and fructose both *in vitro* and *in vivo*. Importantly we have performed control experiments to ensure the phenotype observed upon fructose exposure is not simply due to reduced levels of glucose (Supplementary Figure 5C).

REVIEWERS' COMMENTS

Reviewer #1 (Remarks to the Author):

The authors have thoroughly addressed my criticisms and I believe the revised paper will be a very interesting read for the Nat Commun readership. I am happy to support it for publication.

Reviewer #2 (Remarks to the Author):

The manuscript is significantly improved.